# Modeling saline fluid flow through subglacial channels

Amy Jenson[1,2], Mark Skidmore[3], Lucas Beem[3], Martin Truffer[2], and Scott McCalla[1]

[1]Department of Mathematical Sciences, Montana State University, Bozeman, Montana, USA
[2]Geophysical Institute, University of Alaska Fairbanks, Fairbanks, AK, USA
[3]Department of Earth Sciences, Montana State University, Bozeman, Montana, USA

**Correspondence:** Amy Jenson (ajjenson@alaska.edu)

**Abstract.** Subglacial hydrological systems impact ice dynamics, biological environments, and sediment transport. Previous numerical models of channelized subglacial flow have focused on fresh water in temperate ice, without considering variable fluid chemistry and properties. Saline fluids can exist in cold glacier systems where freshwater cannot, making the routing of these fluids critical for understanding their influence on geochemical and physical processes in relevant glacial environ-
ments. This study advances previous efforts by modeling saline fluid in cold glacier systems, where variable fluid chemistry significantly influences melt rates and drainage processes. We model the drainage of a hypersaline subglacial lake through an ice-walled channel, highlighting the impact of salinity on channel evolution. The model results show that in subglacial systems at salinity-dependent melting points, channel walls grow more slowly when fluids have higher salt concentrations, leading to significantly lower discharge rates. At higher salinities, more energy is required to warm the fluid to the new melting point as
the brine is diluted which reduces the energy available for melting the channel walls. We also highlight the impact of increased fluid density on subglacial drainage and the importance of accounting for accurate suspended sediment concentrations when modeling outburst floods. This model provides a framework to assess the impact of fluid chemistry and properties on the spatial and temporal variation of fluid flux.

## 1 Introduction

Subglacial hydrology is of fundamental importance to the dynamics and evolution of ice masses (Flowers, 2018; Morlighem et al., 2014). The presence, distribution, and geometry of the subglacial water system have direct effects on rates of ice sliding, ice mass flux, erosion, and deposition (e.g. Russell et al., 2006; Bell et al., 2007; Stearns et al., 2008; Siegfried et al., 2016; Larsen and Lamb, 2016; Seroussi et al., 2017; Carrivick and Tweed, 2019; Keisling et al., 2020). The subglacial hydrological system affects the distribution and character of subglacial biological communities and influences water and nutrient flux into surrounding water bodies (e.g. Neal, 2007; Kjeldsen et al., 2014; Meerhoff et al., 2019; Mikucki et al., 2004; Vick-Majors et al., 2020). Whether the goal is to project future sea level rise, understand glacier bed forms, model ocean circulation, or to investigate potential extra-planetary habitats through Earth analogs, we require an understanding of the distribution and dynamics of subglacial hydrological systems (e.g. Nienow et al., 2017; Forte et al., 2016).

Subglacial lakes have been observed to drain episodically through outburst floods and less catastrophically through longer-lived drainage events. There is a significant body of work on modeling the drainage of glacial lakes (e.g. Röthlisberger, 1972;

Nye, 1976; Spring and Hutter, 1981; Fowler, 1999; Clarke, 2003; Evatt et al., 2006; Kingslake, 2015; Schoof, 2020; Jenson et al., 2022). Many subglacial hydrology models have assumed that drainage from a subglacial lake occurs through ice-walled channels at the ice-bed interface (e.g. Nye, 1976; Fowler, 1999; Clarke, 2003; Evatt et al., 2006). Collectively, this work has focused on fresh water at the pressure melting point and neglected the consideration of water chemistry.

The chemistry of the subglacial water influences the character of the hydrological system. Depression of the pressure melting point through increased solute concentration is one potential mechanism to explain the presence of subglacial water in locations with a subglacial temperature below, sometimes significantly below, the pressure melting point (Mikucki et al., 2015). Locations with observable saline discharge occur in both Antarctic and Arctic settings such as Blood Falls, Taylor Glacier, East Antarctica and Borup Fiord Pass Glacier, Ellesmere Island Canada (Trivedi et al., 2018; Lyons et al., 2019). The salinity of the
englacial brine feeding Blood Falls is approximately 125 psu (compared to $\approx 35$ psu for seawater) but the precise geometry of the subglacial brine system beneath Taylor Glacier is not fully understood (Badgeley et al., 2017; Lyons et al., 2019). Hubbard et al. (2004) inferred that a zone 3-6 km upglacier from the terminus contained saturated sediments or ponded water, based on radar data, and widespread hypersaline groundwater has been detected as far as 5.7 km upglacier from the terminus using transient electromagnetic techniques (Mikucki et al., 2015). Hypersaline lakes have been inferred to exist beneath the Devon
Ice Cap, Canadian high Arctic from airborne radio echo sounding data, with predicted salinity in the range of 140 to 160 psu (Rutishauser et al., 2018, 2022), although a more recent study, Killingbeck et al. (2024) using seismic and electromagnetic data argues that the bed is dry or frozen, where the subglacial lakes were inferred. Despite locations of known saline fluid flow in subglacial environments, the effects of increased salinity on the geometry and flow in subglacial hydrological systems remains unknown.

Subglacial water composition is expected to impact the geometry and dynamics of the subglacial hydrological system. For instance, the hydraulic potential field is modified through the density of the fluid; saline fluid can have a significantly different flow path than fresh water for the same glacier geometry (Badgeley et al., 2017; Rutishauser et al., 2022). Channel size and evolution are also expected to differ as the result of fluid chemistry.

An understanding of the impact fluid chemistry has on subglacial systems is important for mapping and classifying sub-
glacial hydrological features using radar. The size, continuity, and electrical conductivity of subglacial channels determines the detectability of subglacial features by radar remote sensing. Constraints provided by modelling inform radar system design decisions such as power requirements, center frequency, and antenna geometry (Scanlan et al., 2022). The ongoing development of multi-polarization radar system and radar processing algorithms increases the detectability of variations in subglacial hydrological organization (Scanlan et al., 2022, 2020). The expected geometry of subglacial features is an important specifi-
cation for the design of radar systems (Scanlan et al., 2022). The response in the geometry of subglacial features to changes in the discharge, position along a flowline, and aqueous chemistry provides constraints for the technological and scientific development of new radar systems.

Basal thermal regimes have been shown to impact the solutes, nutrients, and microbes found in the subglacial systems (Dubnick et al., 2020) and subglacial fluid flow can transport these materials leading to a change in the geomicrobiology of
local and nearby environments (Mikucki et al., 2004). We hypothesize that both the basal thermal regime and solute con-

centrations influence the subglacial hydrological system by altering the effective pressure and fluid flux (which in turn will influence geomicrobiology). A better understanding of the flow dynamics in cold ice is important for characterizing the distinct biogeochemistry in saline subglacial systems.

By modeling saline fluid flow through cold ice, we seek to address the following questions: how significant is the effect of salinity on channel wall melt rates?; how does the salt concentration change along the channel in response to the melting of the channel walls?; and in what systems is the consideration of fluid chemistry and fluid properties important for understanding subglacial hydrology? To answer these questions, we mathematically investigate channel evolution and discharge over time in response to variable fluid chemistry. The results show that the radius of an evolved channel is smaller for saline fluid than the fresh water equivalent when both glacier-lake systems are at their respective salinity and pressure-dependent melting points. The smaller channel cross-sections affect the temporal and spatial evolution of fluid flux for saline fluid. The differences related to fluid chemistry are greatest for high discharge rates which are generated by high volume lakes and channels with steep bed slopes and circular geometry. Additionally, we find that fluid density has a substantial influence on discharge rates which has relevance to suspended sediments and outburst floods.

## 2   Model description

We construct a lake-drainage model in which the water flows from a subglacial lake through an R-channel (Röthlisberger, 1972; Nye, 1976). In contrast with previous approaches, we allow for varying salt concentrations in fluid flowing from a subglacial lake. We follow the implementation of Fowler (1999) and Kingslake and Ng (2013). In particular, the equations describing channel evolution and the conservation of mass, momentum, and energy are identical to those in Fowler (1999) with the differences being (i) the fluid density and the melting point of the fluid are functions of salinity, (ii) the assumptions around fluid temperature are related to salinity, and (iii) an additional equation is needed to solve for salinity along the channel and in time. The model equations from Fowler (1999) are described with our assumptions and in our notation in Sec. 2.1 and changes to the model equations are described in Sec 2.2. For a list of model variables and parameters along with the consistently used parameter values see Table 1.

### 2.1   Model equations

We assume a subglacial conduit on an inclined bed slope beneath ice of constant thickness (Fig. 1). The negative basic hydraulic gradient is the sum of the glacier geometry related terms,

$$\psi = \rho_b g \sin B - \frac{\partial P_i}{\partial s}, \tag{1}$$

where $B$ is the conduit slope (assumed to be constant along the channel and the same as the bed slope), $s$ is the along-flow coordinate parallel to the bed, $\rho_b$ is the density of the fluid, and $g$ is gravitational acceleration (Kingslake and Ng, 2013, Eq. 5). The ice-overburden pressure $P_i$ in [Pa] is given by $P_i = \rho_i g H$ where $H$ is the glacier thickness. Since we assume the ice thickness is constant, the change in ice-overburden pressure along the channel is zero and $\psi = \rho_b g \sin B$.

**Table 1.** List of model parameters and variables. Values of constants are specified in brackets.

| Variable | Description |
| --- | --- |
| $\rho_i, \rho_w, \rho_b$ | densities of ice [917 kg m$^{-3}$], water [997 kg m$^{-3}$], and brine |
| $g$ | gravitational acceleration [9.81 m s$^{-2}$] |
| $\mathcal{L}, \sigma_b$ | latent heat of fusion [$3.34 \times 10^5$ J kg$^{-1}$] and specific heat capacity of brine |
| $A, n$ | ice flow law parameter and exponent [3] |
| $K$ | ice flow parameter for conduit closure |
| $f, \mathcal{R}, n_i, n_b$ | friction factor, hydraulic roughness, and roughness of ice [0.6 m$^{-1/3}$ s] and bed material [0.16 m$^{-1/3}$ s] |
| $x, s$ | horizontal and bed-parallel spatial coordinates |
| $B, L$ | bed slope, channel length |
| $Q, m$ | discharge along the conduit and melt rate |
| $S, r$ | cross-sectional area and channel radius |
| $P_i, \psi$ | ice overburden pressure and basic hydraulic gradient |
| $N$ | effective pressure |
| $h, h_i$ | lake depth and initial lake depth |
| $H$ | ice thickness from surface to bed adjacent lake |
| $V, V_i$ | volume of lake and initial volume of lake |
| $\theta_i, \theta_w, \theta_b$ | temperatures of basal ice, water, and brine |
| $\hat{\theta}_w, \hat{\theta}_b$ | melting point of water and brine at pressure |
| $\hat{\theta}$ | salinity- and pressure-dependent melting point of ice |
| $\beta, \hat{\beta}$ | salt concentration [psu] and [kg m$^{-3}$] respectively |

Assuming an existing channel, the channel walls open due to melt and close due to creep closure. Together these govern the rate of change of the conduit cross-sectional area $S$ with respect to time $t$,

$$\frac{\partial S}{\partial t} = \frac{m}{\rho_i} - KSN^n, \tag{2}$$

where $m$ is the melt rate in [kg m$^{-1}$ s$^{-1}$] (Fowler, 1999, Eq. 2.1). $K = 2A/(n^n)$ is a function of the Glen's flow law parameter $A$ and exponent $n$ (Evatt et al., 2006). The effective pressure $N$ is the difference between the ice-overburden pressure and the water pressure in the channel. We calculate $A$ as a function of ice temperature using the Arrhenius relation and relevant calibrated values (Cuffey and Paterson, 2010, Eqns 3.35 & 3.36).

Mass conservation relates the rate of change of conduit area to the spatial gradient in discharge $Q$, and the production of 100 water due to melt (Fowler, 1999, Eq. 2.2), such that

$$\frac{\partial S}{\partial t} + \frac{\partial Q}{\partial s} = \frac{m}{\rho_w}. \tag{3}$$

We assume turbulent flow and use Manning's formula to empirically relate the total potential hydraulic gradient (negative basic hydraulic gradient and the effective pressure gradient) to friction along the channel. Note that Manning's formula was

derived for fresh water, but due to a lack of empirical data with saline fluid in ice, we use the Manning's friction factor for fresh water in ice as used in Fowler (1999). Regardless of salinity, large uncertainties exist in Manning's formula even in fresh water systems due to the large variability in the value of Manning's friction factor (Pohle et al., 2022). The conservation of momentum equation is then,

$$\psi + \frac{\partial N}{\partial s} = f \rho_b g \frac{Q^2}{S^{8/3}}, \tag{4}$$

where $f$ is the friction factor (Fowler, 1999, Eq. 2.3). For a circular ice-walled channel, $f = (4\pi)^{2/3} \mathcal{R}^2$ and $\mathcal{R} = n_i = 0.06$ m$^{-1/3}$ s where $\mathcal{R}$ is the hydraulic roughness and $n_i$ is the roughness of ice (Clarke, 2003). For a semi-circular channel at the bed, $f = (2(\pi+2)^2\pi)^{2/3}\mathcal{R}^2$ where $\mathcal{R} = \pi/(2+\pi)n_i + (1 - (\pi/(2+\pi))n_b$ and $n_b = 0.16$ m$^{-1/3}$ s is the roughness of the bed material (Fowler, 1999, Eq. 2.24).

The conservation of energy equation is

$$Q\left(\psi + \frac{\partial N}{\partial s}\right) = \rho_b \sigma_b \left(S\frac{\partial \theta_b}{\partial t} + Q\frac{\partial \theta_b}{\partial s}\right) + m\mathcal{L} + m(\theta_b - \hat{\theta}), \tag{5}$$

where $\theta_b$ is the temperature of the brine, $\hat{\theta}$ is the melting point of the ice, $\sigma_b$ is the specific heat capacity of the brine, and $\mathcal{L}$ is the latent heat of fusion for ice (Fowler, 1999, Eq. 2.4). Following Röthlisberger (1972) and Nye (1976), we neglect the heat transfer equation which is equivalent to assuming any heat generated from flow is instantaneously transferred to the channel walls. Consequently, we also need to make an assumption around fluid temperature and the melting point which are related to the salinity of the fluid.

## 2.2 Consideration of brine

As the brine flows from the subglacial lake, any melt occurring at the channel walls will add fresh water and dilute the brine along the channel. We use a partial differential equation to describe the concentration of salt $\hat{\beta}$ [kg m$^{-3}$] at position $s$ and time $t$ in response to changes in channel cross-sectional area and discharge. The fluid is moving along the channel at velocity $v$ which gives the flux of salts per square meter

$$\phi = v\hat{\beta} - D\frac{\partial \hat{\beta}}{\partial s}$$

where $D$ is the diffusion coefficient. The mean velocity of the fluid is given by $v = Q/S$. We calculate a Péclet number of $(Pe) > 10^8$ which suggests advection dominates diffusion in fluid flow and assume diffusion is negligible. With this assumption, the flux of salt moving through a channel cross-section with area $S$ is

$$\phi = \frac{Q\hat{\beta}}{S}.$$

Assuming there is no brine added along the channel and there is no accretion on the channel walls, the salt concentration equation is,

$$\frac{\partial}{\partial t}\left(\hat{\beta}S\right) = \frac{\partial}{\partial s}\left(-Q\hat{\beta}\right). \tag{6}$$

In the case of a semi-circular channel, contact with the ground could be a source for salts in the fluid flow. Although we do not include it here, such a mechanism could be accounted for in our model as a source term in Eq. 6.

The salt concentration $\hat{\beta}$ discussed above is in [kg m$^{-3}$] in order to be compatible with the model. These values for salinity are converted to a standard unit for measuring salinity [psu] before calculating the density and melting point in Eqs. 8 and 9 respectively using the conversion $\hat{\beta}$ [kg m$^{-3}$] $= 1000\beta(\rho_b)^{-1}$ [psu]. The salinity in the lake is constant in time since no fluid is being added to the lake which gives the boundary condition $\hat{\beta}(0,t) = 1000\beta(0,t)(\rho_b)^{-1}$ where $\beta(0,t)$ in [psu] is prescribed at the beginning of the simulation. The dilution of the brine along the channel is minimal so at the beginning of the simulation we assume that the salt concentration in the channel is equal to the concentration in the lake, that is $\hat{\beta}(s,0) = \hat{\beta}(0,t)$.

Fluid properties such as the density, specific heat capacity, and the melting point of ice are functions of salinity ($\beta$ in practical salinity units [psu]). The specific heat capacity of brine [J kg$^{-1}$ °C$^{-1}$] is calculated with the salinity [psu] and temperature of the brine [° C] in the lake following,

$$\sigma_b = 4217.4 - 3.72\theta_b - 7.64\beta \tag{7}$$

using the first order terms from Eqn. A3.11 in Gill (1982).

As salt concentration changes, the density of brine and the melting point of ice also vary spatially and temporally. The density of the brine (in [kg m$-3$]) as a function of salt concentration ($\beta$ in practical salinity units [psu]) under 1 bar using the FREeZing CHEMistry (FREZCHEM) model from Wolfenbarger et al. (2022), disregarding higher order terms, results in

$$\rho_b = 1000 + 0.763\beta. \tag{8}$$

Note we do not account for changes in density due to pressure or temperature. Using the same FREZCHEM model, we calculate the melting point of ice due to salinity and adjust for water pressure. The melting point in [°C] of ice in contact with saline fluid at pressure $P_w$ is

$$\hat{\theta} = -c_b\beta - c_n P_w = -c_b\beta - c_n(P_i - N) \tag{9}$$

where $c_b = 6.05 \times 10^{-2}$ °C psu$^{-1}$ and $c_n = 7.45 \times 10^{-8}$ °C Pa$^{-1}$. We assume the lake and surrounding ice system is in thermal equilibrium which requires that at the lake, the ice and brine temperatures, $\theta_i$ and $\theta_b$ respectively, are equal and at the salinity and pressure-dependent melting point $\hat{\theta}$. For a given salt concentration in the lake, we calculate the melting point at the lake and set the ice and brine temperatures equal to that temperature at the lake. We assume the ice temperature remains constant in time and along the channel.

We assume the brine temperature and the melting point are equal (Röthlisberger, 1972; Werder et al., 2013) and evolve in response to the changes in salinity along the channel and in time. Given that expected changes in the brine temperature are small and we are modeling slow drainage events, we make the assumption that the brine temperature is in quasi-steady state ($\frac{\partial\theta_b}{\partial t} = 0$). Using the assumption that fluid temperature is equal to the melting point ($\theta_b = \hat{\theta}$) along with Eq. (9) gives the simplified conservation of energy,

$$Q\left(\psi + \frac{\partial N}{\partial s}\right) = \rho_b\sigma_b Q\left(-c_b\frac{\partial\beta}{\partial s} + c_n\frac{\partial N}{\partial s}\right) + mL. \tag{10}$$

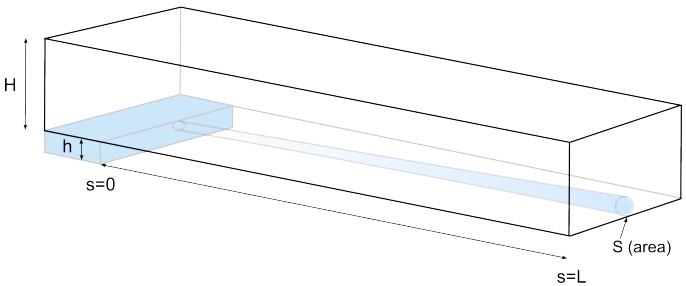

**Figure 1.** Schematic of simple glacier geometry and subglacial hydrological system with a R-channel draining a subglacial lake.

The term on the left hand side of Eq. 5 is the total work done which must be balanced by the sum of the energy (i) needed to raise the brine temperature to the new salinity and pressure-dependent melting point and (ii) lost to melting due to latent heat.

We assume a circular channel for most simulations, but we do compare the effect of circular vs semi-circular channels in Sec. 3. The main differences between these assumptions are that in the semi-circular case (1) the fluid is flowing along the bed and therefore the roughness of the bed must be accounted for instead of the roughness of ice and (2) the substrate may contain

some salts. We do not account for (2) in our model. We do account for (1) through the friction factor which appears in Eq. 4 and changes depending on the channel geometry and roughness of the channel walls or the bed.

### 2.3 Channel boundary conditions

The only fluid flux from the subglacial lake is the brine flowing out of the channel. Thus the rate of change of lake volume $V$ is given by

$$\frac{dV}{dt} = -Q(0,t). \tag{11}$$

We assume a box-shaped lake which gives the lake hypsometry

$$\frac{h}{h_i} = \frac{V}{V_i} \tag{12}$$

where $h$ is the depth of the lake, $h_i$ is the initial lake depth, and $V_i$ is the initial lake volume. For the treatment of more complicated lake geometries, see Kingslake (2013).

Implicitly differentiating gives

$$\frac{dV}{dt} = \frac{dV}{dh}\frac{dh}{dt} = \frac{V_i}{h_i}\frac{dh}{dt} \tag{13}$$

and by substitution, the lake depth evolves with time following

$$\frac{dh}{dt} = \frac{h_i}{V_i}(-Q(0,t)). \tag{14}$$

We assume the lake drains slowly enough that the ice roof drops with the lake depth following Evatt et al. (2006), so as the

lake drops the effective pressure at the lake is still the difference between the ice overburden pressure and the fluid pressure in

the lake. The boundary condition where the conduit meets the lake is $N(0,t) = 0$ (Evatt et al., 2006). We impose a Neumann boundary condition at the end of the channel where

$$\frac{\partial}{\partial s}N(s,t)\bigg|_{s=L} = 0. \tag{15}$$

We choose this boundary condition (opposed to $N = 0$) in order to solve the system numerically in a more efficient way (see Appendix A for details). Neumann boundary conditions on effective pressure at the end of the channel have been used to solve similar systems of equations without an influence on the qualitative results (Kingslake, 2015; Evatt et al., 2006).

## 2.4 Summary of model equations

The full model contains five unknowns ($N, S, m, Q,$ and $\beta$) and five model equations (Eqs. 2, 3, 4, 5, and 6) which are solved simultaneously. The model equations contain the derived variables $\hat{\theta}, \rho_b,$ and $\psi$ which depend on salinity. The model equations written in terms of the salinity-dependent derived variables are listed below.

Channel evolution: $\frac{\partial S}{\partial t} = \frac{m}{\rho_i} - KSN^3$ 
$\qquad$ Conservation of energy: $Q\left(\psi + \frac{\partial N}{\partial s}\right) = \rho_b \sigma_b Q\left(-c_b\frac{\partial \beta}{\partial s} + c_n\frac{\partial N}{\partial s}\right) + mL$

Conservation of mass: $\frac{\partial S}{\partial t} + \frac{\partial Q}{\partial s} = \frac{m}{\rho_b}$ 
$\qquad$ Salt Concentration: $\frac{\partial}{\partial t}\left(\hat{\beta}S\right) = \frac{\partial}{\partial s}\left(-Q\hat{\beta}\right)$

Conservation of momentum: $\psi + \frac{\partial N}{\partial s} = f\rho_b g\frac{Q^2}{S^{8/3}}$

These five equations are non-dimensionalized and solved numerically as described in Appendix A. The system of equations is solved using a constant grid spacing of 50 m and a constant time step of less than 1 second for each simulation (smaller time steps are required for higher salinities). After the solution to the salt concentration equation is obtained at each time and space step, the melting point $\hat{\theta}$ and the density $\rho_b$ are updated along with the basic hydraulic gradient $\psi$, which is a function of density using Eqs. 9, 8, 1 respectively.

## 3 Results

Idealized model simulations were run to investigate the impact of brine on discharge rates, channel radius, effective pressure, and the duration of lake drainage. The model runs until open channel flow occurs, at which point the model run ends. Unless otherwise specified, the parameters used for the baseline simulations are as follows: ice thickness above channel $H = 100$ m, initial lake volume $V_i = 5 \times 10^5$ m$^3$, initial lake depth $h_i = 10$ m, channel length $s = 1000$ m, initial channel radius $r = 0.25$ m, bed and conduit slope $B = 3°$, and circular channel geometry. A range of different values were explored for each parameter listed in Table 2 while holding all other parameters equal to the baseline simulation values.

To investigate the effect of saline fluid, we ran six scenarios with $\beta = \{0, 5, 10, 25, 50, 125 \text{ psu}\}$ to explore the range of possible outcomes. Based on the salinity, we set the ice temperature and initial melting point of the ice and initial brine temperature to $\hat{\theta} = \{-0.06, -0.37, -0.67, -1.58, -3.09, -7.63°\text{C}\}$ respectively. The discharge rates are lower for fluid with higher salt concentrations (Fig. 2a). Higher salt concentrations decreases the peak velocity reached and increases the amount

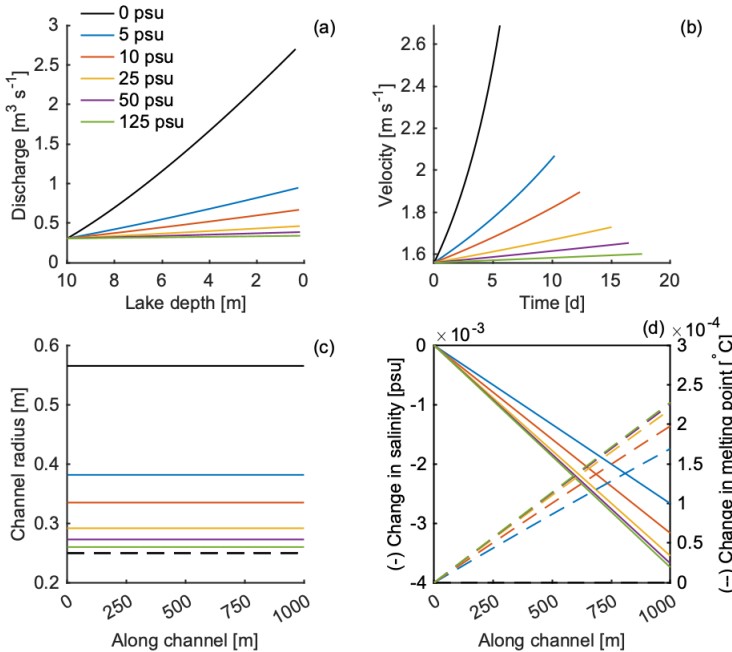

**Figure 2.** The impact of brine on discharge, velocity, channel radius, and changes in salt concentration and melting points for various initial salt concentrations, shown by the colors in (a). (a) Discharge at the lake outlet as the lake drains. (b) Velocity at the lake outlet over time in days. (c) Channel radius along the length of the channel at the time the lake has emptied. The dashed line is the initial channel radius of 0.25 m. (d) The solid lines (associated with the left axis) indicate the difference between the final salt concentration along the channel after the lake has emptied and the initial salt concentrations shown in the legend of (a). The dashed lines (right axis) refers to the difference along the channel between the melting points at the end of the simulation and initial melting points.

of time to reach peak velocity (Fig. 2b). The peak velocity and drainage duration change nonlinearly with increased salt concentrations. That is, a difference in salinity of 5 psu from fresh water has a substantial influence on fluid velocities, while a difference in salinity of the same amount at higher salinities results in much smaller velocity changes (Fig. 2b). After the lake has drained, the channel radius is larger than the initial channel radius for all salinities tested (Fig. 2c). At the end of the simulations, the channel radius is greatest for fresh water, which is more than twice the channel radius of a lake with a salinity

of 125 psu.

    The salt concentration decreases linearly along the channel in all scenarios, with the most significant changes occurring in cases with higher salinities (Fig. 2d). As the salinity decreases, the melting point increases along the channel (Fig. 2d).

    We systematically vary parameters to explore the sensitivity of the model and the impact of channel geometry, lake volume, initial channel radius, and bed slope on discharge and the duration of drainage (Fig. 3). The channel geometry (circular vs.

semi-circular) changes the time of lake drainage, as well as the peak discharge for fresh water and for a lake with salinity of 10 psu (Fig. 3a). For a semi-circular channel, the difference in time until lake drainage is more significant than the difference in

peak discharge between the freshwater and brine scenarios. For the circular channel, the difference in peak discharge is greater than the difference in the duration of time until lake drainage. In both scenarios, the circular channel drains the lake in less than half the time than for a semi-circular channel and the peak discharge is over three times as high when the channels are circular which is partially due to double initial cross-sectional area for those simulations.

The volume of the lake impacts the discharge and the timing of drainage by extending the amount of time the model is run (Fig. 3b). The discharge curves for all lake volumes follow the same curve until the smaller lakes drain. The lake continues to drain for greater lake volumes. For a freshwater lake, the peak discharge roughly triples when comparing $V_i = 1 \times 10^5$ m$^3$ with $V_i = 5 \times 10^5$ m$^3$ and $V_i = 5 \times 10^5$ m$^3$ with $V_i = 1 \times 10^6$ m$^3$, where as for a lake with a salinity of 10 psu the peak discharge approximately doubles. Larger lake volumes extend the time until the lake drains which results in non-linear increases in the differences between fresh water and saline fluid. The differences in timing and peak discharge for saline and freshwater lakes are largest for greater lake volumes (Fig. 3b).

The initial channel radius impacts the amount of time until the lake drains and the peak discharge (Fig. 3c). With a salinity of 10 psu and an initial channel radius of 0.2 m, the lake drains substantially slower ($\approx 10$ days) compared to an initial channel radius of 0.3 m. For each 0.1 m increase in initial channel radius, the peak discharge increases non-linearly for both saline fluid and freshwater, but the rate of increase is less pronounced for saline fluid. Increasing the bed and channel slope increases the peak discharge and decreases the time to reach that peak (Fig. 3d). We varied fresh water ($\beta = 0$ psu) and brine ($\beta = 10$ psu) along with the channel slope and found that for lower slopes there is a larger difference in timing between brine and fresh water and for greater slope there is a larger difference in peak discharge between brine and fresh water. Higher bed slopes lead to higher discharge rates more quickly (Fig. 3d). For all choices of parameters, we find that brine decreases the discharge rates and increases the time until lake drainage compared to fresh water.

We explored lake depths of $5 - 15$ m, ice thicknesses of $100 - 1000$ m, and channel lengths of $100 - 5000$ m and found that these parameters do not substantially impact the results for the parameter combinations described (data not shown). For all initial conditions on effective pressure, the effective pressure along the channel tends towards zero over time. For a list of parameters and the range of values explored, see Table 2.

As the walls of the channel melt and the brine is diluted, the density of the brine is not constant along the channel. We have accounted for this in our simulations, but neglecting these changes does not have a substantial influence on the results because the changes are very small (data not shown).

## 4 Discussion

The results of this model suggest that the consideration of brine in relevant glacial systems is important for capturing the dynamics of drainage through ice-walled channels: a failure to consider salt even for low salinities leads to substantially different estimates on channel formation and drainage rates and timescales (Fig. 2). The consideration of salinity is more important when considering systems with high discharge rates (high lake volume and steep bed slopes, see Fig. 3b,d). Large

**Table 2.** Prescribed model parameter variables and descriptions with baseline simulation values and ranges explored.

| Variable | Description | Baseline | Range |
|---|---|---|---|
| $s$ | length of channel | 1000 m | $[500 - 5000 \text{ m}]$ |
| $B$ | bed (conduit) slope | $3°$ | $[2 - 4°]$ |
| $r$ | initial channel radius | 0.25 m | $[0.1 - 0.5 \text{ m}]$ |
| $h$ | initial lake depth | 10 m | $[5 - 15 \text{ m}]$ |
| $H$ | ice thickness above bed and channel | 100 m | $[100 - 1000 \text{ m}]$ |
| $V_i$ | reference volume of lake | $5 \times 10^5 \text{ m}^3$ | $[1 \times 10^5 - 1 \times 10^6 \text{ m}^3]$ |
| $\beta$ | salt concentration of brine in lake | 0, 10 psu | $[0 - 125 \text{ psu}]$ |

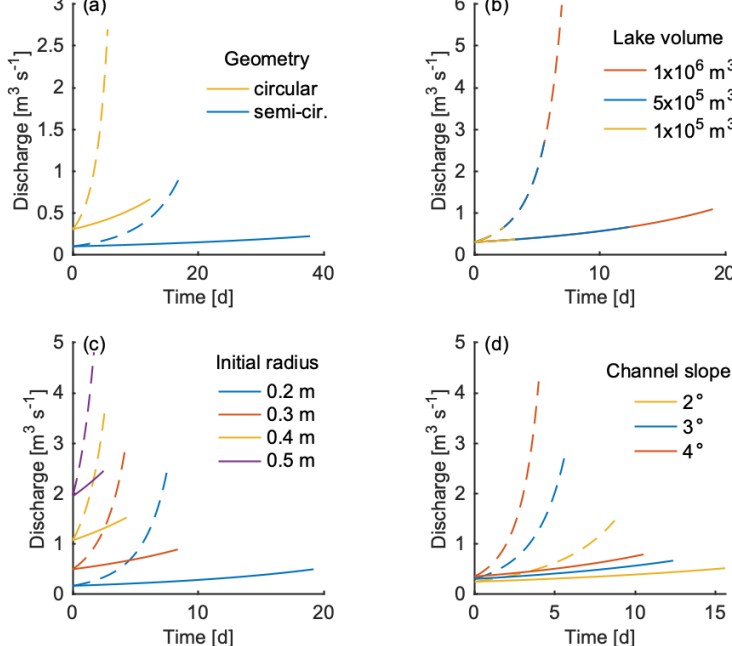

**Figure 3.** Discharge over time for different channel geometries, lake volumes, initial channel radius, and channel (bed) slope for lakes with fresh water (dashed lines) and 10 psu (solid lines). (a) Channels with semi-circular geometries are denoted in blue while circular geometries are in yellow. (b) Lake volumes for $V_i = 1 \times 10^6 \text{ m}^3$ (red line), $V_i = 5 \times 10^5 \text{ m}^3$ (blue line), and $V_i = 1 \times 10^5 \text{ m}^3$ (yellow line) are plotted over top of one another. (c) The initial channel radius is varied such that the blue line is $r = 0.2$ m, the red line is $r = 0.3$ m, the yellow line is $r = 0.4$ m, and the purple line is $r = 0.5$ m. (d) Channel slope is varied to $2°$ (in yellow), $3°$ (in blue), and $4°$ (in red).

circular channels also tend to lead to more substantial differences in the fluid dynamics between high and low concentrations
of salt (Fig. 3a,c).

## 4.1 Effects of salinity

The presence of salt in fluid decreases the melting point and increases the density of the fluid, both of which have implications
for how fluid drains through an R-channel from a subglacial lake. An obvious impact is that saline fluid can remain liquid
at subzero temperatures and a subglacial lake system can exist below the pressure melting point. The differences related to
the salinity-dependent melting point also influence how the channel grows in time. The presence of salt in the system tends to
decrease the amount of energy available for melting the channel walls because energy is needed to change the brine temperature
to the increasing melting point.

For freshwater systems, a positive feedback allows for dynamic channel growth where higher discharge rates generate more
energy for melting, opening the channels, and allowing for even higher discharge rates. This feedback is the mechanism
responsible for developing an efficient drainage system with large channels and for glacier lake outburst floods. A system with
saline fluid at a sub-zero temperature reduces this positive feedback. The melting of the channel walls results in an increase in
the melting point of ice due to the changes in salinity after the addition of fresh meltwater. While melting of the channel wall
increases the cross-sectional area and therefore the discharge, more energy is now required to increase the brine temperature to
the new melting point and less energy is available for melting.

The ratio of fresh meltwater from the channel walls to saline discharge from the lake determines brine concentration. A
channel with a smaller cross-sectional area has a higher surface-area-to-volume ratio compared to a larger channel, leading
to a higher ratio of meltwater to saline discharge and larger changes in brine concentration along the channel. Channels are
smaller for higher salinities (Fig. 2c). Brine concentration and the resulting changes in the melting point vary linearly along
the channel, with larger gradients observed at higher salinities (Fig. 2d). Larger spatial gradients in brine concentration require
more energy to raise the brine temperature to the new salinity and pressure-dependent melting point along the channel. This
reduces the energy available for melting, which strongly inhibits rapid channel growth (Eq. 10). The effect of inhibited channel
growth is largest at higher salinities, where the initial brine temperature is lowest ($-7.63°$ C). However, the system is more
sensitive at lower salinities, where even small increases in salinities lead to significant changes, as the initial brine temperature
is close to $0°$ C ($-0.37$ to $-0.67°$ C) (Fig. 2c). Inhibited channel growth due to salinity results in more gradual increases in
velocity over time (Fig. 2b).

Density impacts channel growth in the opposite way. A fluid with a higher salt concentration has a higher density. As a denser
material moves through a gravitational potential, more energy is generated and available for melting. In Fig. 4a, a temperate
freshwater system ($\hat{\theta} = -0.06°C$) is modeled with the density of fresh water and a fluid density of $\rho = 1098 \, \text{kg m}^{-3}$, equivalent
to the density of brine with a salinity of 125 psu. The fluid with the higher density results in a higher peak discharge (Fig. 4a).

Modeling all other changes related to salinity (including the treatment of the melting point and fluid temperature) while
holding the density of the brine constant and equal to that of fresh water ($\rho = 1000 \, \text{kg m}^{-3}$) results in the discharge rates
shown in Fig. 4b. Without the consideration of accurate brine densities, there is almost no difference in the peak discharge even

for the highest salinities with the highest densities (Fig. 4b). Higher fluid densities amplify the dynamic feedback leading to channel growth and higher discharge rates (Fig. 4a). However, when the channel growth is reduced by the effects of salinity, sub-zero initial brine temperatures, and thus changes in the melting point, this dynamic feedback does not occur and the influence of density is limited. In part, this is due to the fact that density changes almost linearly with salinity (Eq. 8) while the impact of salinity on discharge is non-linear.

It is important to note that the increased density generates more energy for melting only when the flow is gravity driven (i.e., a downward sloped inclined channel). For example, if the flow of a saline fluid is driven uphill by glacial overburden pressure, the fluid velocity will be slower compared to the fluid velocity of fresh water.

Due to the substantial influence of changes in the melting point on energy available for melting, the considerations of thermodynamics and assumptions around the brine temperature have large impacts on the results when modeling saline fluid in channelized systems. Assuming constant brine temperature (opposed to assuming the brine temperature remains equal to the melting point as we do here), results in substantially different results, where the influences of density become the first order effect. However, this assumption is likely unrealistic and differs from the convention in previous studies. In other models of subglacial channel flow that do not explicitly include temperature, the temperature is set to follow the pressure melting point of the ice walls within the channel (e.g. Röthlisberger, 1972; Werder et al., 2013).

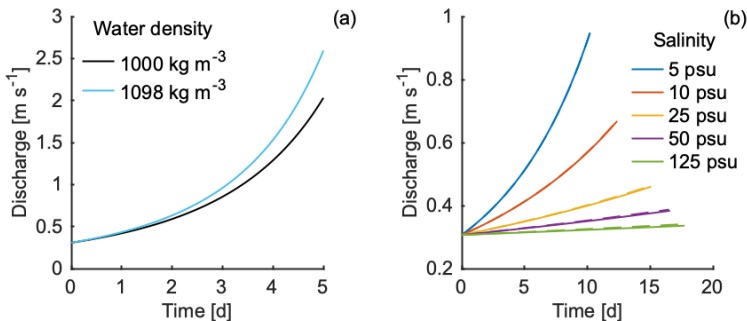

**Figure 4.** (a) Discharge curves for drainage from a temperate freshwater system with densities equivalent to that of fresh water and brine with a salinity of 125 psu. (b) Discharge curves for brine with $\beta = 5, 10, 25, 50, 125$ psu, but all with constant density equal to fresh water (1000 kg m$^{-3}$). Simulations from the full model, where both density and the effects of saline fluid at subzero temperatures are considered, are plotted in the dashed lines which are hardly visible.

## 4.2 Drainage types of saline systems

There are limited observations of saline outflows and subglacial lake drainage events which makes it difficult to understand the hydrological state of such systems. Each drainage system has a unique combination of (i) source and concentration of

salt, (ii) brine and ice temperature, (iii) bed topography and glacier geometry determining the hydraulic potential, and (iv) the mechanism initiating drainage which all play a role in the state of the subglacial system.

Our model results show that when the brine temperature remains equal to the melting point, the sub-zero saline fluid reduces the positive feedback that leads to the rapid channel growth and increases in fluid velocity. Additionally, our results show that effective pressures along the entire channel tend towards zero over time, that is, the modeled channel is a high water pressure system. Consistent high water pressure and reduced channel growth may suggest that saline systems do not tend toward channelization, and may exist as distributed systems. However, further modeling of a distributed system with sub-zero saline fluids would be required to characterize such behavior.

Alternatively, Badgeley et al. (2017) hypothesize that the mechanism of drainage at Blood Falls is not by means of opening and closing of R-channels, but by brine injection into basal crevasses. More observational and modeling work is needed to understand more how saline fluid behaves in a spatially connected subglacial hydrologic system.

## 4.3 Implications for outburst floods

As shown in Sec. 4.1, density can have a significant effect on fluid flow regardless of salinity. Outburst floods often result in disproportionate amounts of suspended sediment which increases the density of the water (Snorrason et al., 2002; Church, 1972). Discharge from outburst floods are typically on the order of $100 - 1000$ m$^3$ s$^{-1}$ (Walder and Costa, 1996, Table 1) and can contain suspended sediment concentrations ($SSC$) of 70.7 g L$^{-1}$ (Beecroft, 1983; Old et al., 2005) and in some extreme cases over $400$ g L$^{-1}$ (Maizels, 1997). The density of sediments $\rho_s$ depends on the rock type and clast size, but typically range from $2350 - 2760$ kg m$^{-3}$ (Frederick et al., 2016; Guan et al., 2015; Chikita, 2004). The combined fluid density $\rho_c$ of water with suspended sediments is related to suspended sediment concentration by,

$$\rho_c = \rho_w + (\rho_s - \rho_w)\frac{SSC}{\rho_s}. \tag{16}$$

We model an outburst flood from a subglacial lake with fresh water at $0°$ C and a volume of $V_i = 1\text{x}10^7$ m$^3$ with all other parameters equal to those in the baseline simulation (Table 2). We vary the suspended sediment concentrations from $SSC = \{0, 63, 125, 188, 251\}$ g L$^{-1}$ to arrive at the combined fluid densities of $\rho_c = \{1000, 1040, 1080, 1120, 1160\}$ kg m$^{-3}$ to simulate different suspended sediment loading (for $\rho_s = 2760$ kg m$^{-3}$). There is a significant difference between the peak discharge of floods with a lower density fluid ($Q \approx 115$ m$^3$ s$^{-1}$) than with a higher density fluid ($Q \approx 140$ m$^3$ s$^{-1}$) as well as the timing of the peak discharge (Fig. 5). Neglecting to account for sediment loading and accurate water densities could lead to inaccurate results when modeling outburst floods where gravity assists in driving fluid flow. The influence of density is particularly important for systems with high gravitational potential energy and high fluid density which together can significantly increase fluid flow rate.

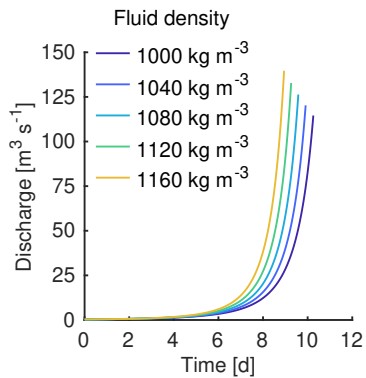

**Figure 5.** Outburst flood hydrographs for water with different suspended sediment concentrations and therefore varying fluid densities.

## 5  Conclusions

We have presented a subglacial hydrology model which includes the consideration of saline fluid. Salt allows fluid to exist below the freezing point of fresh water and increases the density of the fluid. We show that if a channel exists, hypersaline fluid can flow through an ice-walled channel when the brine and ice are at the salinity and pressure-dependent melting point. Our results suggest that a higher salt concentration and commensurate lower brine temperature decreases the peak discharge and increases the time for a fixed volume to drain from a subglacial lake.

The main driver of the decreased discharge rates for higher salinities is the influence of changes in the salinity-dependent melting point. As the salinity along the channel changes due to melting of the channel walls, the melting point changes and consequently the energy needed to melt the channel walls. While more energy is generated when a higher density fluid moves through a gravitational potential, this increase in energy for higher salinities is minimal compared to the energy needed to raise the brine temperature to the melting point.

This study shows that accounting for fluid properties is crucial for accurately modeling subglacial hydrology in relevant systems (i.e., saline fluid, more or less dense fluid). For a lake with a salinity of 125 psu, which is approximately the measured value at Blood Falls, and an initial brine temperature of $-7.63°$ C, the peak velocity reached is $40\%$ lower and the lake drains $9\%$ slower than for a freshwater lake. The duration of drainage is most sensitive to initial channel radius and channel geometry while peak discharge is most sensitive to lake volume, channel slope, and channel geometry (circular vs. semi-circular). We explored the influence of varying fluid density related to suspended sediment loads on outburst floods and found that peak discharge is significantly higher for a high density fluid ($22\%$ higher than pure water when the fluid density is $1160$ m$^3$ s$^{-1}$). In our model, subzero saline fluid reduced channel growth which may have implications for the type of subglacial drainage system (channelized vs distributed) that forms with such fluids.

We make a number of simplifying assumptions in the model and use arbitrary parameters for ice thickness, lake volume, channel length, bed slope, and initial channel radius due to a lack of available data on subglacial hypersaline systems. We also impose brine temperatures that remain equal to the ice melting point. In this light, the model presented here should be viewed as an initial exploration of the impact of brine on the dynamics of a subglacial hydrological system. Additional modeling efforts are needed to provide a thorough sensitivity and stability analysis. Further research is required to understand the initiation of

drainage in cold saline environments and the influence of fluid density on drainage networks and outburst floods.

*Code availability.* MATLAB script files for full model are available at https://doi.org/10.5281/zenodo.10775488 (Jenson et al., 2024).

## Appendix A: Numerics

After non-dimensionalization, the model equations to be solved numerically are the following.

**Dimensionless model equations**

Channel Evolution:
$$\frac{\partial S}{\partial t} = Q\left(\psi + \gamma_1 \frac{\partial N}{\partial s} + \gamma_2 \frac{\partial \beta}{\partial s}\right) - SN^3, \tag{A1}$$

Conservation of Mass:
$$\frac{\partial Q}{\partial s} = \epsilon\left(rm - \frac{\partial S}{\partial t}\right), \tag{A2}$$

Conservation of Momentum:
$$\frac{\partial N}{\partial s} = \frac{1}{\delta}\left(\frac{Q^2}{S^{8/3}} - \psi\right), \tag{A3}$$

Salt Concentration:
$$\lambda \frac{\partial}{\partial t}\left(\hat{\beta}S\right) = \frac{\partial}{\partial s}\left(-\hat{\beta}Q\right), \tag{A4}$$

Boundary Conditions: $N(0,t) = 0,$ (A5)

$$\frac{\partial}{\partial s}N(s,t)\bigg|_{s=1} = 0, \tag{A6}$$

$$\hat{\beta}(0,t) = \hat{\beta}(0,0) \tag{A7}$$

$$\frac{dh}{dt} = \zeta Q(0,t). \tag{A8}$$

Model and Scaling Parameters:

$$N_0 = (Kt_0)^{-1/3}, \quad m_0 = \frac{Q_0\psi_0}{\mathcal{L}}, \quad S_0 = \left(\frac{f\rho_b g Q_0^2}{\psi_0}\right)^{3/8}, \quad t_0 = \frac{\rho_i S_0}{m_0},$$

$$r = \frac{\rho_i}{\rho_b}, \quad \epsilon = \frac{s_0 m_0}{Q_0\rho_i}, \quad \delta = \frac{N_0}{s_0\psi_0}, \quad \gamma_1 = \frac{N_0\rho_b\sigma_b c_n}{s_0\psi_0}, \quad \gamma_2 = \frac{\beta_0\rho_b\sigma_b c_b}{s_0}, \quad \lambda = \frac{S_0 s_0}{t_0 Q_0}, \quad \zeta = \frac{t_0 h_i Q_0}{pV_i h_0}$$

We use the subscripts $j = 1, 2, ...n$ to denote the grid points along the channel which are separated by $\Delta s$ and the superscripts $i = 1, 2, ...m$ denote time steps separated by $\Delta t$. Note, the conservation of energy equation has been substituted into the channel evolution equation to give Eq. (A1).

To solve Eq. (A8), we follow Kingslake (2013) in using the Forward Euler Method to evolve the lake depth forward in time.

$h^{i+1} = h^i + \Delta t \zeta Q_1^i.$

Similarly, we solve Eq. (A1) using the same method. The channel cross-sectional area $S$ is moved forward in time at all grid points by

$$S_j^{i+1} = S_j^i + \Delta t \left( Q_j^i \left( \psi + \gamma_1 \frac{N_j^i - N_{j-1}^i}{\Delta s} + \gamma_2 \frac{\beta_j^i - \beta_{j-1}^i}{\Delta s} \right) - S_j^i (N_j^i)^3 \right)$$

for $j = 1, 2, ... n$.

To evolve these two equations forward in time, the discharge and effective pressure at time step $i$ is needed. These variables can be found simultaneously solving the mass and momentum equations.

We follow Fowler (1999) and Kingslake (2013) in assuming $\epsilon$ is small enough to neglect the terms containing $\epsilon$ in Eq. (A2). With parameter values, $m_0$, $s_0 = 1000$ m, and $\rho_i = 917$ kg m$^{-3}$, $\epsilon$ is on the order of $10^{-3}$ and thus we neglect these terms which simplifies Eq. (A2) to

$$\frac{\partial Q}{\partial s} = 0. \tag{A9}$$

This is equivalent to assuming that any melt generated along the channel is small in comparison to the volume of fluid moving through the channel from the lake. Solving Eq. (A3) for the discharge $Q$ and evaluating at the end of the channel gives

$$Q^2(1,t) = \left( \delta \frac{\partial N}{\partial s}(1,t) + \psi(1,t) \right) S(1,t)^{8/3}. \tag{A10}$$

From Eq. (A6) and the assumption that the discharge is always positive (flowing out of the lake),

$$Q(1,t) = \sqrt{S(1,t)^{8/3} \psi(1,t)}. \tag{A11}$$

From Eqs. (A11) and (A9), $Q(s,t) = Q(1,t)$ and we arrive at the following equation for discharge as a function of time

$$Q(t) = \sqrt{S(1,t)^{8/3} \psi(1,t))}.$$

We solve this equation by calculating the discharge profile at each grid point by

$$Q_j^i = \sqrt{(S_n^i)^{8/3} \psi_n^i}. \tag{A12}$$

To calculate the effective pressure along the channel, we start with the boundary condition at the lake given in Eq. (A5) and use Eq. (A3) to iterate

$$N_{j+1}^i = N_j^i + \frac{\Delta s}{\delta} \left( \frac{(Q_j^i)^2}{(S_j^i)^{8/3}} - \psi_j^i \right) \tag{A13}$$

from $j = 1, 2, ... n$.

To solve the dimensionless brine equation Eq. (A4), we use an upwind difference scheme such that

$$\hat{\beta}_j^{i+1} = \hat{\beta}_j^i - \Delta t \left( \frac{\beta_j^i}{S_j^i} \left( \frac{S_j^i - S_j^{i-1}}{\Delta t} \right) + \frac{Q_j^i}{\lambda S_j^i} \left( \frac{\hat{\beta}_j^i - \hat{\beta}_{j-1}^i}{\Delta s} \right) \right).$$

After calculating the non-dimensional salt concentration, we re-dimensionalize the salt concentration to be in units of [kg m$^{-3}$] and convert this to [psu] using,

$$\beta_j^{i+1} = \frac{\hat{\beta}_j^{i+1}\hat{\beta}_0 1000}{\rho_{b j}^i}. \tag{A14}$$

The density of brine along the channel can be updated with the new salt concentration using Eq. 8. Similarly the basic hydraulic gradient is updated using the new brine density. In order to solve the system, initial conditions are needed for cross-sectional area, effective pressure, discharge, salinity along the channel. At the beginning of the simulation, we assume a constant cross-sectional area and salt concentration along the channel. We use the initial cross-sectional area at the end of the channel to calculate the initial discharge curve along the channel using Eq. A12. We assume the initial effective pressure profile linearly increases from $N = 0$ at the lake to $N = P_i$ at the end of the channel.

*Author contributions.* This publication is a direct result of AJ's Master's Thesis who was lead on all aspects of the research. AJ, SM, and MS conceived the study and contributed to the research. LB and MT helped with clarifications of the methods. All authors contributed to editing the paper.

*Competing interests.* The authors declare that they have no conflict of interest.

*Acknowledgements.* We thank Natalie Wolfenbarger for providing equations for the density and melting point as functions of salinity using FREEZCHEM. We acknowledge Jack Dockery for the helpful discussions regarding the derivation of the salinity equation. We would also like to thank two anonymous reviewers for their comments which have greatly improved this manuscript. MS and LB are supported by NASA award 80NSSC20K1134. SGM would like to acknowledge the support of NSF grant nos. 1813654, 2112085 and the Army Research Office (W911NF-19-1-0288).

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
