# Peer review of "Modeling saline fluid flow through subglacial channels"

_EGUsphere, 2023_

## Referee Comment (RC1)

**Review: "Modeling saline fluid flow through subglacial ice-walled channels and the impact of density on fluid flux"**

by Jenson et al.

Submitted to *Cryosphere*

**1   General**

In this paper, the authors analyze flow of brine through subglacial channels. Classic analysis for subglacial channels, e.g. Röthlisberger (1972), neglects the role of solutes in the subglacial flow, which is unlikely to be the case in the natural environment. The authors present the first analysis, as far as I know, of the role of solutes in the subglacial hydrologic system. I found the paper to be interesting to read, yet a bit thin and confusing in places. I was surprised to learn that the major effect is the density of the water, rather than the melting point depression, but that leads me to wonder about the model construction. I will likely support publication but would be interested in one or two rounds of revision to clarify some of my questions.

**2   Remarks**

1. I am a bit skeptical that the main effect is the density change, given that the salt can so effectively lower the melting point. My rationalization is that the water is already at the local melting point (even in the pure water case) and the salt would just allow channels to exist at lower water temperatures. However, the ice temperature will be warmer, suggesting that the salt can melt the ice and there will be heat transfer. I think this is what is captured by equation (7) but I don't yet fully understand.

2. In the summary of model equations, I am not sure that there are enough equations listed to close the system. It seems like the 7 unknowns are $N$, $S$, $m$, $\psi$, $m$, $Q$, $\beta$, and $\hat{\theta}$ and there are 5 equations listed. The other two are the statement of $\psi$ and the melting point $\hat{\theta}$, which are described earlier, but I think it would be clearer if they were stated here as well. Also, I think it would be useful to include a statement of the boundary conditions and initial conditions in this section. As a counting exercise, it would be useful to see all of the conditions required.

**3   Specific comments**

A few small things that I thought of:

1. A self citation is fair in the section on outburst floods.

2. equation (1): the assumptions here imply that $\psi = \rho_b g \sin(B)$, is that correct? If not, what is $\psi$?

3. line 101: "is therefore non-constant" could be "therefore varies"

4. line 184: I think it would be useful to add the units after the list of $\beta$ values.

5. line 288: I am confused about the effective pressure values coming out of the model: are they all negative and extremely small? I would expect some in the kPa range, rather than something like $-100$ Pa. Am I missing something?

6. line 306: the authors state that they are using 'somewhat arbitrary' parameters. Which ones specifically? And why? Would it be better to add a citation of possible better values?

7. In the appendix, where the dimensionless model is written out, what are the boundary conditions on $\beta$? I don't think they are stated and would be useful.

8. I think some of the model development that is included in the appendix could be useful to put back into the main text.

9. I am not a fan of the notation $Q(s = 1, t)$ since it looks cluttered (e.g. equation A9) and it is an odd statement. I prefer $Q$ at $s = 1$ or $Q(1, t)$, if absolutely necessary.

**References**

H. Röthlisberger. Water pressure in intra- and subglacial channels. *J. Glaciol.*, 11(62): 177–203, 1972. doi: 10.3198/1972JoG11-62-177-203.

---

## Author Comment (AC1)

**Response to Anonymous Referee #1**

Thank you for your thoughtful feedback that will help us improve our manuscript.

**General**

In this paper, the authors analyze flow of brine through subglacial channels. Classic analysis for subglacial channels, e.g. R¨othlisberger (1972), neglects the role of solutes in the subglacial flow, which is unlikely to be the case in the natural environment. The authors present the first analysis, as far as I know, of the role of solutes in the subglacial hydrologic system. I found the paper to be interesting to read, yet a bit thin and confusing in places. I was surprised to learn that the major effect is the density of the water, rather than the melting point depression, but that leads me to wonder about the model construction. I will likely support publication but would be interested in one or two rounds of revision to clarify some of my questions.

We have responded to each of your comments below.

**Remarks**

1. I am a bit skeptical that the main effect is the density change, given that the salt can so effectively lower the melting point. My rationalization is that the water is already at the local melting point (even in the pure water case) and the salt would just allow channels to exist at lower water temperatures. However, the ice temperature will be warmer, suggesting that the salt can melt the ice and there will be heat transfer. I think this is what is captured by equation (7) but I don't yet fully understand.

We agree that the main effect of a saline fluid is the lowering of the melting point. If a hypersaline fluid was circulated through a glacier that is at the local melting point for freshwater, this would have interesting consequences and density may not be the dominant effect. However, in this manuscript we are considering the case where initially the ice is at the melting point of the saline fluid. We will add a statement in the Introduction to make this clearer (see below, line 63). We do state this in the conclusion (lines 292-296),"Salt allows fluid to exist below the freezing point of freshwater and increases the density of the fluid. We show that if a channel exists, hypersaline fluid can flow through ice-walled channels at the same subzero temperature. Our results suggest that higher salt concentration increases the peak discharge and decreases the duration of time for a fixed volume to drain from a subglacial lake. The main driver of this difference is the higher fluid density." While we do claim that the main effect is a result of a higher density fluid, this is under the assumption that the ice and liquid are in thermal equilibrium. Thermal equilibrium is what we would expect in a subglacial lake system so

we set the ice and saline fluid temperature to be the same at the beginning of each simulation. The saline fluid in a subglacial lake will not be colder than the surrounding ice temperatures. Given the different salinities and therefore assumed ice/brine temperatures, the main impact of an increase in salt concentration is through the increased density of the fluid. We will clarify that the influence of density is only under the assumption of equal temperatures at the melting point by making the changes listed below.

Line 8:
"The model results show that given a subglacial system at the salinity-dependent melting point, channel walls grow more quickly when fluid contains higher salt concentrations which lead to higher discharge rates. We show this is due to a higher density fluid moving through a gravitational potential which generates more energy for melting."

Line 63:
"The results show that the fluid flux is greater with saline fluid than the fresh water equivalent when both glacier-lake systems are at their respective salinity and pressure-dependent melting points. The larger channel cross-sections affect the temporal and spatial evolution of fluid flux for saline fluid."

Line 233:
"Modeling all other changes related to salinity (including the treatment of the melting point and fluid temperature) while holding the density of the brine constant and equal to that of fresh water results in the discharge rates shown in Fig. 3b."

Line 293:
"We show that if a channel exists, hypersaline fluid can flow through an ice-walled channel when the brine and ice are at the salinity and pressure-dependent melting point."

Line 295:

"The main driver of the increased discharge rates as a function of salinity is the higher fluid density associated with higher salt concentrations. More energy is generated and available for melting the channel walls when a higher density fluid moves through a gravitational potential. While some of this energy is used to warm the ice to the new pressure melting point down the channel as the brine is diluted by meltwater, the melt is minimal compared to the discharge from the lake and therefore does not impact the discharge rates. Aside from the influence of salinity on the depression of the melting

point, the greatest difference on fluid flux when considering saline fluids is related to the change in density."

2. In the summary of model equations, I am not sure that there are enough equations listed to close the system. It seems like the 7 unknowns are N, S, m, ψ, m, Q, β, and ˆθ and there are 5 equations listed. The other two are the statement of ψ and the melting point ˆθ, which are described earlier, but I think it would be clearer if they were stated here as well. Also, I think it would be useful to include a statement of the boundary conditions and initial conditions in this section. As a counting exercise, it would be useful to see all of the conditions required.

You are correct that there are five model equations, which are solved simultaneously. The model contains 5 unknowns (N, S, m, Q, β) which are represented by the 5 model equations (Eq. 3, 4, 5, 7, and 8). The model derived variables ˆθ, ρ, and ψ are updated spatially and temporally at each space and time step after the updated salt concentration β is known using Eqs. 1 and 6, and the equation for ˆθ listed in line 114. We note that we do eliminate the equation one unknown (m) and one equation (Eq. 7) after non-dimensionalizing the system and making the assumptions discussed in Section 4.4. However, the number of unknowns and equations always remain the same.

We realize now that the number of unknowns is not stated and we are not clear in distinguishing between variables, derived variables, equations, and parameters. We will be more accurate and consistent in our language by making the changes listed below.

1. We will change the first sentence of the caption in Table 1 to say "List of model parameters and variables."
2. We will also change line 73 to "For a list of model variables and parameters along with the consistently used parameter values see Table 1".
3. Additionally, we will change line 168 to read, "The full model contains five unknowns (N, S, m, Q, β) and five model equations (Eq. 3, 4, 5, 7, and 8) which are solved simultaneously. The model equations contain the derived variables ˆθ, ρ_b, and ψ which depend on salinity. The model equations written in terms of the salinity-dependent derived variables are listed below." We will add an equation number to reference in line 114.

**Specific comments**

A few small things that I thought of:
1. A self citation is fair in the section on outburst floods.

We will add a citation for Jenson et al. (2022) to line 24.

2. equation (1): the assumptions here imply that $\psi = \rho_b g \sin(B)$, is that correct? If not, what is $\psi$? 1

Yes, that is correct. We will clarify this by adding to line 79, "...the change in ice-overburden pressure along the channel is zero and $\psi = \rho\_b\ g\ \sin(B)$."

3. line 101: "is therefore non-constant" could be "therefore varies"

Agreed.

4. line 184: I think it would be useful to add the units after the list of $\beta$ values.

We agree and the text has been updated to "beta = {0, 50, 100, 150, 200 psu}".

5. line 288: I am confused about the effective pressure values coming out of the model: are they all negative and extremely small? I would expect some in the kPa range, rather than something like −100 Pa. Am I missing something?

Yes, the effective pressures are all essentially zero, which implies that the water pressure remains high throughout the channel. A common boundary condition at the end of the channel is N = 0, which is approximately what we see here even with the Neumann boundary condition. The reason the values are very slightly negative is due (i) the cross-sectional area is smallest at the end of the channel which is related to the additional energy needed to melt the channel walls after the brine is diluted and the melting point increases and (ii) the density of the fluid is less at the end of the channel which is also a result of the dilution of the brine. Although the water pressures are slightly higher than ice overburden pressure, we believe this qualitatively does not affect the results. We will add at line 288, "The effective pressure at the end of the channel for all simulations is extremely close to zero (−200 < N ≤ 0 Pa) and therefore we claim that the sign of the effective pressure is negligible and does not qualitatively affect our results."

6. line 306: the authors state that they are using 'somewhat arbitrary' parameters. Which ones specifically? And why? Would it be better to add a citation of possible better values?

Thanks for this comment. We agree this is a vague statement. We will change this to say, "We make a number of simplifying assumptions in the model and use arbitrary

parameters for ice thickness, lake volume, channel length, bed slope, and initial channel radius due to a lack of available data on subglacial hypersaline systems."

7. In the appendix, where the dimensionless model is written out, what are the boundary conditions on β? I don't think they are stated and would be useful.

We discuss the imposed boundary conditions on β in lines 147-150, but we agree that they should be listed here as well.

Your comment has pointed out to us that we have used $\beta\_0$ in both the non-dimensionalization and as the salinity of the lake (in lines 147-150). We will change lines 147-150 to read, "The salinity in the lake is constant in time since no fluid is being added to the lake which gives the boundary condition $\hat{\beta}(0,t) = \beta(0,t) * 1000/\rho\_b$ where $\beta(0,t)$ is in [psu] is prescribed at the beginning of the simulation. When the cross-sectional area of the channel is small, there is less melting and the dilution of the brine is minimal so at the beginning of the simulation we assume that the salt concentration in channel is equal to the concentration in the lake, that is $\hat{\beta}(s,0) = \hat{\beta}(0,t)$."

8. I think some of the model development that is included in the appendix could be useful to put back into the main text.

From information previously stated in the appendix, we will add to the end of the model development section, "The system of equations are solved using a constant time step of approximately 3 seconds and a constant grid spacing of 20 m. After the solution to the salt concentration equation is obtained at each time and space step, the melting point $\hat{\theta}$, and the density $\rho_b$ are updated along with the basic hydraulic gradient $\psi$, which is a function of density using Eqs. 7, 6, 1 respectively."

9. I am not a fan of the notation Q(s = 1, t) since it looks cluttered (e.g. equation A9) and it is an odd statement. I prefer Q at s = 1 or Q(1, t), if absolutely necessary.

Okay, we will change this notation to Q(1, t). We will also change all other places in the manuscript such as in Section 4.4 where the same convention is used.

**References**

H. R¨othlisberger. Water pressure in intra- and subglacial channels. J. Glaciol., 11(62): 177–203, 1972. doi: 10.3198/1972JoG11-62-177-203.

---

## Author Comment (AC2)

**Response to Anonymous Referee #2**

Thank you for your thorough feedback that will help us improve our manuscript.

This paper describes a jökulhlaup model, based on R-channel drainage, taking into account effects of saline water on the dynamics. It finds that the impact of salinity on melt-rates is limited and the biggest impact is due to the increased density of the fluid on hydraulics.

This short paper has interesting findings and is novel as no-one has looked into impacts of salinity on R-channel dynamics. It has however a few shortcomings which need to be rectified before publication.

We have responded to each of your comments below.

Major comments

==============

The somewhat surprising take-home (that the main difference compared to freshwater discharge is due to the higher density and not due to higher melt rates) should be elaborated a bit more. I would state the following, possibly in the Abstract & Conclusion:

- salinity makes liquid water possible at sub-zero temperatures: fresh water would be frozen. So this is really the biggest difference: liquid water versus no liquid water.

- once the equations are solved at the sub-zero temperature given by the melting point of the saline solution, it is actually pretty obvious that impact on melt is minimal: dilution of the water as it traverses the R-channel is minimal as melt is really small compared to discharge. Even for a setting where there is much more potential energy available, this statement holds. Thus it makes sense that density has the biggest impact.

A few comments in response:

1. We agree that the main impact of salinity is the depression of the melting point and we will explicitly make that point in the manuscript, as suggested (see below, especially l. 63)
2. Once we assume that the melting point is depressed and the ice and brine are at that temperature, the main difference compared to freshwater is due to higher density. The greater density of saline fluid does **result in higher melt rates** due

to the additional energy generated. We will make this more clear in the Abstract and Conclusion.

3. The dilution of the brine would have the opposite impact than what we see in our results. If the brine was diluted more, then the melting point would increase (as a result of lower salinity) and less melt would occur which would result in lower discharge rates. You are correct that the melt is minimal compared to the discharge and this impacts the results. We will clarify this in the conclusion.

In response to these concerns (and similar concerns from Referee #1), we will make the changes shown below.

Line 8:
"The model results show that given a subglacial system at the salinity-dependent melting point, channel walls grow more quickly when fluid contains higher salt concentrations which lead to higher discharge rates. We show this is due to a higher density fluid moving through a gravitational potential which generates more energy for melting."

Line 63:
"The results show that the fluid flux is greater with saline fluid than the fresh water equivalent when both glacier-lake systems are at their respective salinity and pressure-dependent melting points. The larger channel cross-sections affect the temporal and spatial evolution of fluid flux for saline fluid."

Line 233:
"Modeling all other changes related to salinity (including the treatment of the melting point and fluid temperature) while holding the density of the brine constant and equal to that of fresh water results in the discharge rates shown in Fig. 3b."

Line 293:
"We show that if a channel exists, hypersaline fluid can flow through an ice-walled channel when the brine and ice are at the salinity and pressure-dependent melting point."

Line 295:

"The main driver of the increased discharge rates as a function of salinity is the higher fluid density associated with higher salt concentrations. More energy is generated and available for melting the channel walls when a higher density fluid moves through a gravitational potential. While some of this energy is used to warm the ice to the new pressure melting point down the channel as the brine is diluted by meltwater, the melt is

minimal compared to the discharge from the lake and therefore does not impact the discharge rates. Aside from the influence of salinity on the depression of the melting point, the greatest difference on fluid flux when considering saline fluids is related to the change in density."

Provide a sketch (as fig. 1) of the setup for an overview, that way also the coordinate system and orientation are defined.

Okay, thank you. We will include the following schematic as Figure 1.

[Figure]

Figure 1. Schematic of simple glacier geometry and subglacial hydrological system with a R-channel draining a subglacial lake.

With many of the equations in section 2 I struggle:

- Eq6 and the following three unnumbered eqs have some issues, see line-by-line comments

- my understanding is that the authors assume that water-temp equal to the local melting point as stated on l72 (i.e. temperature is dictated by pressure and salinity). Thus \theta, \theta_i and \theta_b are equal and hence the last term in Eq7 is zero. If temperature was treated as a free variable, as in Fowler 1999, then this term would be needed.

You are correct that \theta_hat, \theta_i and \theta_b are all equal at the beginning of the simulations. However, because the brine is diluted as freshwater is added, the melting point theta_hat decreases along the channel. Therefore, the last term in Eq. 7 is non-zero except (i) along the channel at time zero and (ii) where the channel meets the lake throughout the simulation. The terms on the right hand side in Eq. 7 are identical to those presented in Fowler (1999), but the variables differ in that the brine temperature is

assumed constant and the melting point is allowed to vary in space and time as a result of the changing salt concentration along the channel. We will clarify this by deleting the sentence in line 115 and adding the following discussion of the assumptions. "We assume the lake and surrounding ice system is in thermal equilibrium which requires that at the lake, the ice and brine temperatures, θi and θb respectively, are equal and at the salinity and pressure-dependent melting point ^θ. For a given salt concentration in the lake, we calculate the melting point at the lake and set the ice and brine temperatures equal to that temperature. We assume the ice and brine temperatures remain constant in time and along the channel; this is realistic for most freshwater systems (Clarke, 2003). However, the melting point only remains constant at the lake and evolves in response to the changes in salinity along the channel and in time."

  --> this whole temperature treatment is confusing.  I had a quick look at the code and I think that temperature is indeed not a free variable.

You are correct that ice and brine temperatures are parameters and not free variables. They are constant in space and time (except in Section 4.3). However, the melting point \theta_hat is calculated along the channel after each time step given the updated salt concentration, beta. So \theta_hat is a derived variable that depends on the free variable \beta. It was pointed out by Referee 1 as well that it is not clear what are free variables. In order to address this, we will change line 168 to read, "The full model contains five unknowns (N, S, m, Q, β) and five model equations (Eq. 3, 4, 5, 7, and 8) which are solved simultaneously. The model equations contain the derived variables ^θ, ρ_b, and ψ which depend on salinity. The model equations written in terms of the salinity-dependent derived variables are listed below."

- the melt rate $m$ can be solved for in Eq7.  However, I do not understand what is going on here.  As stated above, the last term should ==0.  Also there should be a pressure dependent term to capture the pressure melting point effects.  At least when a linear relation between temperature and pressure is assumed then it takes the form $rho_w c_t  c_t dP_w/ds*Q$ (e.g. Röthlisberger 1972, Eq 17); maybe the authors try to capture this with their last term of Eq7, but I don't think that can be done like this.  A term in similar spirit would need to be added for the salinity dependent effects, presumably featuring a $d\beta/ds$.  In fact, this is the quintessential equation to be stated as that is the novel one, all others are known.

The term you are referring to (rho_w c_t  c_t dP_w/ds*Q) is described in Eq. 3 in Röthlisberger 1972 as the energy needed for the change in water temperature to the new pressure melting point. This is because in Röthlisberger's model, (i) the water temperature always remains at the pressure melting point and (ii) there is a change in

ice overburden pressure which results in a change in the melting point. However, in our model we assume (i) the fluid temperature remains constant and (ii) the ice thickness is constant. Because the melting point increases as a result of the brine dilution, some energy is needed to raise the ice temperature to the new pressure melting point before melting, which is accounted for in the last term of Eq. 7.

The brine temperatures in our simulations are considered constant, except in Section 4.3 where we explore a small change in temperature equal to the change in the melting point. In this section in Eq. 15, we include a term to account for the energy needed to the change in brine temperature. In this term ($\sigma \ \partial\hat{\theta}/\partial t \ \rho \ S$), we account for the dbeta/ds term you mentioned, although it is hidden in $\hat{\theta}$. The melting point is a function of salinity and pressure, but since the ice thickness is constant there is no change in pressure along the channel (dP_i/ds = 0) and the only change in the melting point is related to the change in the salt concentration (dbeta/ds). It is written slightly differently than in Röthlisberger 1972 but it is essentially the same. In our equation the derivative is with respect to time and multiplied by S (m^3) instead in his model where there is a derivative with respect to space multiplied by Q (m^3/s), but it is referring to the same quantity of water per time.

- How Eq15, also featuring the melt rate, then ties in with this is also not clear to me.

In Section 4.3, we assume that the brine temperature remains equal to the melting point (instead of the ice temperature). Therefore, there is some energy needed to raise the temperature of the brine, so Eqn 15 replaces Eqn 7 for this section.

The lake has only ever a simple geometry, thus I would recommend to just keep it simple and shorten that part. A simple "refer to Kingslake 2015 for a treatment of more complicated lake hypsometries."

Okay, thank you. We will remove the dimensionless parameter p to simplify Eqs. 10 11 and 12 and change line 155 to say, "We assume a box-shaped lake which gives the lake hypsometry

(Eq. 10),

where h is the depth of the lake, hi is the initial lake depth, and Vi is the initial lake volume. For the treatment of more complicated lake geometries, see Kingslake (2013)."

Line by line

============

12: nice Introduction

Thank you.

13: (any many others): I prefer incomplete lists of references to be prefaced by a "e.g.".
Many instances in the Introduction (l.15, 18, 24, 26, etc.)

Okay, we will add in lines 15, 18, 21, 24, 26.

30: I would prefer to use an SI units throughout instead of "psu".  At least give a
translation (1psu=1kg/m3).

We agree that remaining consistent would be preferred. However, the units of salinity
are not often referred to in salt concentrations of kg/m3 as is required in our model for
units to agree. Therefore, to be consistent with other literature we present and discuss
the salinity and the melting point and density equations in psu. We will add the
conversion to line 147 by saying, "The salt concentration beta_hat discussed above is in
[kg/m^3] in order to be compatible with the model. These values for salinity are
converted to a standard unit for measuring salinity [psu] before calculating the density
and melting point in Eqs. 6  and (line 110) respectively using the conversion beta_hat
[kg/m^3] = beta * 1000/ rho_b  [psu]. " We will add an equation number to the equation
in line 110 to reference. We will also add the definition of the psu (practical salinity units)
to line 104.

70: Cite the original too, Röthlisberger (1972).  I would write something like: "We
construct a lake-drainage model in which the water flows through a subglacial channel
(Röthlisberger, 1972; Nye, 1976).  We follow the implementation and notation of Fowler
(1999) and Kingslake (2015).  We assume..."  I also like the term "R-channel" and would
use that, but no strong feelings here.

Thank you. We will replace the first two sentences with "We construct a lake-drainage
model in which the water flows from a subglacial lake through an R-channel
(Röthlisberger, 1972; Nye, 1976). We follow the implementation and notation of Fowler
(1999) and Kingslake (2015). In our model, we assume a subglacial conduit on an
inclined bed slope beneath ice of constant thickness (Fig. 1)." where Fig. 1 is the
schematic you recommended.

74: I disagree with Fowler (1999) here and thus with this manuscript: the given definition
is the "negative basic hydraulic potential".  I think it would be good to state the
"negative" somewhere.  Similarly Eq2 gives the negative potential gradient.

Okay, we change both of the definitions to 'negative'.

95: the "(instead of Darcy-Weisbach)" I find confusing as D-W is also for freshwater. Just delete?

Okay, we have deleted "(instead of Darcy-Weisbach)" from this line.

97: I think 0.6m^{-1/3}s should be 0.06 (according to the code).  Also cite where these values are from (maybe Clarke 2003?) as Fowler 1999 does not provide those values.

Yes, you are correct. Thanks for catching this. It should be ni = 0.06 and this value is from Clarke (2003). We will change this and add the citation.

Eq6 and following: I don't think those formulas should be stated with so many digits.  At most two significant digits are needed.  Also state that for the $\rho_b$ and $\Delta\theta_P$ the 3rd and 2nd order terms are not important.  Also, Eq6 is written with \beta as an argument, the others are without argument: should be consistent.

Agreed. We will reduce the equations to include only two significant digits.

Beta can be up to 200 psu so the second and third order terms in Eq. 6 and the equation in line 110 are not completely negligible.

We will be consistent by removing the argument (beta) from Eq. 6.

We have placed the equations in lines 110, 112, and 114 with one equation to describe the melting point where the pressure dependence is linear. Please see response to later comment for the full change.

Eq6: is this correct?  I evaluate $\rho_b(0)=0.1$ but fresh water has a density of 1000kg/m3.

Thank you for catching this. The current equation is presented in g/cm^3 even though we state that it is in kg/m^3. We will convert this to kg/m^3 by multiplying each term by 1000 (as was already done in the code).

Eq6 / I109: I don't understand how the melting point depression can have a constant term.  That would mean that it is always depressed!  Similarly for the change in melting-point due to pressure.  Also, I guess these formula depend on temperature being given in deg-Celsius, that should be stated.

Both of the melting point depression equations are a result of the FREZCHEM model which "...provides access to virtual laboratories where freezing experiments can be conducted for saline water on Earth…" (Wolfenbarger et al., 2022). See Wolfenbarger et al., 2022 referenced in the manuscript for more details. These equations were derived

from fitting a line (with a constant term in the fitting equation) through virtual experimental data with ice at a very large range of pressures, which is not relevant here. Therefore, we will use a new simplified equation that is a function of both salinity and pressure that has been derived to best fit ice pressures where ice thickness is less than 500m and that does not allow for a constant term. We have rerun all simulations and confirmed that changing these equations does not impact our results.

We will replace lines 107-114 with "Using the same FREZCHEM model, we calculate the melting point of ice due to salinity and adjust for the ice-overburden pressure (Chang et al., 2022). The melting point in [∘C] of ice in contact with saline fluid at pressure Pi in [Pa] is

$$\hat{\theta} = -5.81 \times 10^{-7}\beta^3 + 1.24 \times 10^{-6}\beta^2 - 6.05 \times 10^{-2}\beta - 7.45 \times 10^{-8}Pi.$$"

Your comment has pointed out to us that we have made a mistake in the notation. We have denoted $P\_i$ in both bars (in line 113) and in pascal (in line 77). To address this, we will change the melting point equation so that $P\_i$ is in Pa to be consistent.

123: cite Röthlisberger here too, drop Kingslake.

Okay.

132: why suddenly salinity in kgm^-3 instead of "psu"?

The empirical equations from FREZCHEM are given in psu (which is approximately equal to g/kg and ppt) which are standard ways to present salinities. However, in order for the units of salt concentration to match our system of equations we must convert this to kg/m^3. The distinction between the salt concentrations with different units are beta in psu and beta_hat in kg/m^3. Although we acknowledge that this is slightly confusing, we argue that presenting the salinities and related empirical equations in familiar units is important. As mentioned earlier, in order to help address this confusion we have added, "The salt concentration beta_hat discussed above is in [kg/m^3] in order to be compatible with the model. These values for salinity are converted to a standard unit for measuring salinity [psu] before calculating the density and melting point in Eqs. 6 and (line 110) respectively using the conversion beta_hat [kg/m^3] = beta * 1000/ rho_b [psu]. " to line 150.

139-143: this can be abbreviated to just state Eq 8. The reader does not need to see a derivation of a conservation equation.

Okay, we replace lines 139-143 with, "Assuming there is no brine added along the channel and there is no accretion on the channel walls, the salt concentration equation is…"

115: doesn't line 72 state that these temperatures are always equal?

No, the ice and brine temperatures are only equal to the melting point at time zero. While the ice and brine temperatures remain constant, the melting point does not. The melting point changes in response to the change in the salinity according to the equation shown in line 110. This seems to be unclear in the manuscript and we see that this has led to confusion about the treatment of the melting point and energy equation. We hope that the changes made to line 115 mentioned earlier makes this more clear.

167: I wouldn't call this a "no flux" boundary condition as there is water flux through the boundary. Just call it a Neumann BC. Also, what does it mean? What is meant with "We do not require the channel to exit at a terminus or end subaerially"? Where does it end? Or at least where does the simulation end and how do we know that there dN/ds=0?

We will change the language from "no flux" to "Neumann". What we mean by this statement is that the channel may continue on outside of the model domain and the end of the modeled channel is not necessarily at the glacier terminus. There are no other boundary conditions imposed that suggest that the channel does or doesn't end here. We allow for the possibility that we are only modeling a portion of the lake/channel system and do not enforce a water pressure at the end of the channel but allow it to evolve according to the Neumann boundary condition. However, this is not the reason we chose this BC so we will omit this statement and make the following changes at line 165.

"We impose a Neumann boundary condition at the end of the channel where

(Eq. 13).

We choose this boundary condition (opposed to N = 0) in order to solve the system numerically in a more efficient way (see Section 5.4 and Appendix A). Neumann boundary conditions on effective pressure at the end of the channel have been used to solve similar systems of equations without an influence on the qualitative results (Kingslake, 2015; Evatt et al., 2006)."

174: reference the suggested Fig.1 (the sketch) here.

We will add a reference to Fig. 1 in line 71.

Fig.1: could the temperature as a function of distance also be plotted? Maybe even the different components, i.e. press-melt-point term and salinity-melt-point term.

We have added the change in the melting point from the beginning of the simulation to the end along the channel to Figure 1d. The depression of the melting point from pressure is constant (since the ice thickness is constant) so the change in the melting point is only due to changes in salinity. Please see the updated version of Figure 1 below.

[Figure]

Figure 1: The impact of brine on discharge, velocity, channel radius, salt concentration, and effective pressure for various initial salt concentrations, shown by the colors in (a). (a) Discharge at the lake outlet as the lake drains. (b) Velocity at the lake outlet over time in days. (c) Channel radius along the length of the channel at the time the lake has emptied. (d) The solid lines are associated with the left axis which is the difference between the final salt concentration after the lake has emptied and the initial salt concentrations shown in the legend of (a) along the channel. The right axis (dashed lines) refers to the difference between the melting points at the end of the simulations along the channel and the initial melting point, where the change in the melting point is only due to the change in the salinity (shown in the left axis).

230: I think this is vice versa.

This sentence is stated in a confusing way. Yes, there is more energy available for melting when there are higher salt concentrations which is why we see higher discharge rates. However, energy is needed to raise the temperature of the ice surrounding the channel to the new melting point as the melting point increases with the dilution of the brine which is the point we were trying to make here. We will change this sentence to say, "The presence of salt in the system tends to increase the amount of energy needed to melt the channel walls because the ice temperature must increase to the evolving melting point before melting. As the salinity along the channel changes, the melting point changes and subsequently the energy needed to melt the channel walls, although this change is minimal as seen in Fig. 1d."

242: what are "hyperconcentrated sediment concentrations"?  What is this "threshold"?

This phrase is redundant and the sentence is perhaps not worded well. We follow the language used in Maizels (1997) which distinguishes between high sediment concentrations and hyperconcentrated flow. We will reword this to say, "can contain suspended sediment concentrations SSC of up to 70.7 g L−1 (Beecroft, 1983; Old et al., 2005) and in some extreme cases over 400 g L−1 (Maizels, 1997)."

253-259: I recommend to delete these lines as they don't add much

Okay.

288: I concur with Reviewer 1: these N values do not make sense to me.  Is this caused by the Neumann BC on N?  Note that a, in my opinion, more natural and standard BC N=p_i (or p_w=0) leads to dN/ds not equal zero. See for instance in Röthlisberger (1972): Fig.5d shows a setup similar to here and shows p_w -> 0 at the outlet.

We agree that a standard boundary condition to impose at the end of the channels is pw = 0. Another standard boundary condition to impose is N=0 at the terminus which is approximately what we observe in our model, despite the Neumann Boundary condition. The values are very slightly negative due to the dynamics discussed in Section 4.3, but are essentially zero. We will add at line 288, "The effective pressure at the end of the channel for all simulations is extremely close to zero (−200 < N ≤ 0 Pa) and therefore we claim that the sign of the effective pressure is negligible and does not qualitatively affect our results."

A related note: This model does not handle open channel flow and therefore we do expect higher water pressures all the way to the end of the channel.

312: Excellent that the code is provided, thanks!  Ideally the authors should also provide a third script that, when run, will reproduce and write to disc all figures used in the publication.

We only provided the main model here for simplicity. However, we agree all files would be useful. We will add all versions of model code (Section 4.2, 4.3, etc.), the script that runs all the simulations at once, and the scripts that create the figures.

---

## Referee Report (RR1)

**Review of Jenson et al 2023 – Round 2**

Anonymous reviewer 2

August 30, 2023

This is my second review of this manuscript (MS). This paper describes a jökulhlaup model of a brine-filled lake draining through a R-channel taking into account effects of saline water on the dynamics. It finds that the biggest impact of the salinity on R-channel dynamics is due to the increased density of the fluid.

This short paper has interesting findings and is novel as no-one has looked into impacts of salinity on R-channel dynamics.

After clarifications from the authors in the last round of reviews, I am pretty certain that the presented mathematical model of energy conservation is wrong. This needs to be fixed before a publication is possible. I suspect that the fixed model will lead to similar conclusions and that a publication is warranted.

**1 Major comments**

The energy conservation of R-channels is modelled with varying level of complexity:

I the more complex models take water temperature as a state variable and then resolve the heat transfer to the channel walls via some empirical relation (e.g. Nye, 1976; Fowler, 1999; Clarke, 2003). (Note that Fowler (1999) actually ignores pressure melting point effects.)

II the explicit dependence on temperature can be removed by assuming the water temperature follows the pressure melting point of the ice walls of the channel (e.g. Röthlisberger, 1972; Werder et al., 2013)

III to simplify matters further, it can be assumed that the melting point does not depend on pressure (e.g. Schoof, 2010; Hewitt, 2011; Kingslake, 2015).

Of note for this review is that (to my knowledge) all R-channel models which take pressure melting point effects into account use the water pressure to set the melting point and not the ice overburden pressure as assumed in this MS. The reason that the water pressure is used, is because the pressure felt by the ice at the channel walls is the water pressure.

The MS is aiming to make a category II model (with salinity effects added) but makes two, in my opinion, wrong choices:

- Eq. 7 states that the pressure melting term is dependent on the ice pressure $P_i$. Instead (as I just argued above), it should be the water pressure, i.e. the last term should read $-7.45 \times 10^{-8}(P_i - N)$ (with effective pressure $N$). Note that this means that the melting point and also water and ice temperature (they are all equal in a category II model) will in general change in time and space as pressure (and salinity) changes.

- Eq. 8 contains no terms related to the change in salinity and pressure along the channel.

To derive an Eq. 8-like equation, I would do the following steps. Take Eq. 2.4 of Fowler (1999) (on which the MS is based already), which using the MS's notation translates into

$$\rho_w \sigma_i (S \frac{\partial \theta_w}{\partial t} + Q \frac{\partial \theta_w}{\partial s}) = Q(\psi + \frac{\partial N}{\partial s}) - m\mathcal{L} - m\sigma_i(\theta_w - \hat{\theta}). \tag{1}$$

Now assuming a quasi steady-state $\frac{\partial \theta_w}{\partial t} = 0$ and a water temperature at pressure melting point $\theta_w = \hat{\theta}$. To simplify the algebra a little, I drop the higher-order terms of Eq. 7 (but they could be retained), Eq. 7 then reads

$$\hat{\theta}(\beta, N) = \theta_w = -6.05 \times 10^{-2}\beta - 7.45 \times 10^{-8}(P_i - N). \tag{2}$$

Now the $\frac{\partial \theta_w}{\partial s}$ term can be stated as

$$\frac{\partial \theta_w}{\partial s} = \frac{\partial \theta_w}{\partial \beta}\frac{\partial \beta}{\partial s} + \frac{\partial \theta_w}{\partial N}\frac{\partial N}{\partial s} = -6.05 \times 10^{-2}\frac{\partial \beta}{\partial s} + 7.45 \times 10^{-8}\frac{\partial N}{\partial s}. \tag{3}$$

Thus the Eq. 8-like equation reads

$$Q\left(\psi + \frac{\partial N}{\partial s}\right) + \rho_w \sigma_i Q\left(c_\beta \frac{\partial \beta}{\partial s} - c_N \frac{\partial N}{\partial s}\right) = m\mathcal{L} \tag{4}$$

with $c_\beta = 6.05 \times 10^{-2}$ and $c_N = 7.45 \times 10^{-8}$. So, for $\frac{\partial N}{\partial s} > 0$ (i.e. water pressure dropping along flow) melt $m$ is decreased as pressure-melting point increases (see e.g. Röthlisberger (1972)); similarly for $\frac{\partial \beta}{\partial s} > 0$ melt is increased as salinity-melting point decreases. This is how I would expect the model to behave.

Last, note that the behaviour of above equation cannot be captured by the MS's Eq. 8 (even with corrected Eq. 7). I suggest to update both the MS and the numerical code to reflect above. If there is something I am missing in the approach presented in the MS, then it should be well explained as it does not follow the usual approach. (But even then, I would recommend that the MS follows the standard approach to keep it comparable to existing work.)

Also, I recommend to have both pressure-melting point and salinity-melting point effects included in the model and to look at their relative importance.

If the authors agree with this change, then Section 4.3 (in particular Eq. 16) needs to be adjusted as well (or remove the Section).

**2 Line by line comments**

3: "temperate ice"

4: maybe "cold glacier" the "cold-based" suggests to me that only the base of the glacier is cold

Table 1: "melting point of water and brine at pressure"

Eq.4: I think it should be $m/\rho_w$ as the ice melted will produce fresh water.

98: "in ice as used in Fowler"

101: "(Clarke, 2003)"

111: "density due to pressure or temperature."

Eq.7: see above

118: As outlined in "Major comments", this assumption is not how things are normally setup. For instance the cited Clarke (2003) uses the standard approach of setting the ice-wall temperature dependent on pressure (his Eq. 13).

Eq.8: see above

123: should be "$\mathcal{L}$"

137: this equation does not describe a flux as flux should have units of kg/s (or m3/s). Nonetheless Eq. 9 is correct.

Eq.11: remove unnecessary brackets

326: maybe state that the conservation of energy equation is substituted into A1

359: remove extraneous bracket

367: the $M$ is undefined

**References**

Clarke, G. K. C. (2003). "Hydraulics of Subglacial Outburst Floods: New Insights from the Spring–Hutter Formulation". In: *Journal of Glaciology* 49.165, pp. 299–313. DOI: 10.3189/172756503781830728.

Fowler, A. C. (1999). "Breaking the Seal at Grímsvötn, Iceland". In: *Journal of Glaciology* 45.151, pp. 506–516. DOI: 10.3189/S0022143000001362.

Hewitt, I. J. (2011). "Modelling Distributed and Channelized Subglacial Drainage: The Spacing of Channels". In: *Journal of Glaciology* 57.202, pp. 302–314. DOI: 10.3189/002214311796405951.

Kingslake, J. (2015). "Chaotic Dynamics of a Glaciohydraulic Model". In: *Journal of Glaciology* 61.227, pp. 493–502. DOI: 10.3189/2015JoG14J208.

Nye, J. F. (1976). "Water Flow in Glaciers: Jökulhlaups, Tunnels and Veins". In: *Journal of Glaciology* 17.76, pp. 181–207. DOI: 10.3189/S002214300001354X.

Röthlisberger, H. (1972). "Water Pressure in Intra– and Subglacial Channels".
    In: *Journal of Glaciology* 11.62, pp. 177–203. DOI: 10.3189/S0022143000022188.
Schoof, C. (2010). "Ice-Sheet Acceleration Driven by Melt Supply Variability".
    In: *Nature* 468.7325, pp. 803–806. DOI: 10.1038/nature09618.
Werder, M. A. et al. (2013). "Modeling Channelized and Distributed Subglacial
    Drainage in Two Dimensions". In: *Journal of Geophysical Research: Earth
    Surface* 118.4, pp. 2140–2158. DOI: 10.1002/jgrf.20146.

---

## Author Response (AR2)

**Response to Anonymous Referee #2 (Round 2)**

Thank you for your thorough review and suggestions that have helped us substantially improve our manuscript. We have implemented all of the changes you have recommended. Namely, we have changed Eq. 7 as you have outlined in your comments to be a function of salinity and the water pressure (not effective pressure).

We have also made the changes you have suggested to the energy equation (Eq. 8). We now assume that the brine temperature is equal to the melting point (not the ice temperature) along the channel and in time. This has had major implications for the results of the model.

Essentially, by changing the assumption that the brine temperature is equal to the melting point, when the channel walls melt the brine is diluted which increases the melting point. Energy is used to warm the brine to the new melting point which results in less energy available for melting the channel walls. Consequently, salinity has the opposite effect of our original statement and fresh water has a higher peak discharge than saline fluid. The original conclusion about fluid density substantially increasing fluid flux is still true, however this is only the case for freshwater. When considering saline fluid, the resulting increase in energy due density is small compared to the influence of changes in the brine temperature.

This has led to substantial changes to the results and discussion sections of the manuscript.

We have addressed all of the line by line comments exactly as you have recommended.

---

## Author Response (AR3)

**Response to Anonymous Referee #2 (Round 3)**

Dear Referee,

We would like to thank you for carefully reviewing our manuscript. Your input has greatly improved our model and manuscript. We appreciate the time you have invested in multiple reviews. Below we respond to your comments point-by-point.

Best Regards,
On behalf of the authors, Amy Jenson

This paper describes an R-channel model which takes into account the effects of saline water. Its main finding is that an R-channel conducting saline water at the local pressure and salinity melting point will open less fast that an R-channel conducting fresh water. This is because as the channel walls melt, salinity drops and with that the melting point increases. Thus some energy will be needed to heat up the water to the new melting point and thus is not available to melt the channel wall. Note that this is similar to the effect of the pressure dependence of the melting point of ice on R-channel dynamics for fresh water R-channels.

This is the third review for this paper I am doing and I am happy in how this progressed. I think the model is now correct, however, I am not 100% sure about the results. In particular the ones presented in Fig2d seem odd to me: a difference of 10^-5 C between brine and freshwater runs, as shown in Fig2d right axis, is a tiny difference compared to the difference in channel radius or discharge between those runs (Fig2a,c). To get some more context for the amount of melt and thus change in salinity consider the following back of the envelope calculation.

Thank you for noticing the error in Fig 2d. You are correct that the change in salinity (and consequently changes in temperature) should be higher. An error was found in the code that generates Fig. 2d (remnant from debugging). The first time steps were cut off and this is when the majority of the change in salinity occurs.

The total potential energy available is approximately 150m*rho_w*g (100m of ice thickness and about 50m from the 3-degree slope). Taking a typical discharge of 1m3/s the total melt along the channel is approximately (ignoring pressure and salinity effects, i.e. an upper estimate)

$Q * H * rho\_w * g / (rho\_w L) \sim 1 * 150 * 1000 * 10 / (1000 * 3.3e5) \sim 0.004$ m3/s

thus the relative salinity change should be about 0.004/Q ~ 0.004. This is about 3 orders of magnitude larger than what is plotted in Fig2d. Maybe my above estimate is wrong or I do not understand what is plotted in Fig2d. Therefore this needs to be either explained better, in particular how such minuscule changes could have such a big effect, or the calculation needs to be revisited.

After correcting the error in the figure code, the changes in salinity from the end of the simulation from the initial salinity is ~0.004 psu which is close to the back of the envelope calculation you provide. Please see the corrected Fig. 2d.

We have updated the accompanying text describing these results (lines 210 - 212) to "The salt concentration decreases linearly along the channel in all scenarios, with the most significant changes occurring in cases with higher salinities (Fig. 2d). As the salinity decreases, the melting point increases along the channel (Fig. 2d)."

Additionally, what was previously lines 263-269 is now lines 274-283 and reads "The ratio of fresh meltwater from the channel walls to the saline discharge from the lake determines brine concentration. A channel with a smaller cross-sectional area has a higher surface-area-to-volume ratio compared to a larger channel, leading to a higher ratio of meltwater to saline discharge and larger changes in brine concentration along the channel. Channels are smaller for higher salinities (Fig. 2c). Brine concentration and the resulting changes in the melting point vary linearly along the channel, with larger gradients observed at higher salinities (Fig. 2d). Larger spatial gradients in brine concentration require more energy to raise the brine temperature to the new salinity and pressure-dependent melting point along the channel. This reduces the energy available for melting, which strongly inhibits rapid channel growth (Eq. 10). The effect of inhibited channel growth is largest at higher salinities, where the initial brine temperature is lowest (-7.63 ◦ C). However, the system is more sensitive at lower salinities, where even small increases in salinities lead to significant changes, as the initial brine temperature is close to 0◦C (−0.37 to − 0.67◦ C) (Fig.2d). Inhibited channel growth due to salinity results in more gradual increases in velocity over time (Fig. 2b)."

Similarly, the change in temperature along the channel is at most 10^-5C according to Fig2d. However, I would expect a change in temperature due to pressure melt point effects alone of c_n * 100m * rho_w * g ~ 0.07C as the water de-pressurizes from the ice overburden pressure at the lake to zero at the outlet. Again many orders of magnitude difference between this estimate and Fig2d.

As for the change in temperature due to the pressure melting point, there are a few differences between your calculation and our assumptions. We assume the effective

pressure is zero at the lake (following Evatt et al., 2006) and we do not assume the effective pressure is zero at the end of the channel (a result of the setup and numerical method we choose). Instead we require that dN/ds = 0, however it does tend to zero at the end of the channel. Consequently, changes in effective pressure along the channel are small and do not substantially impact changes in the melting point.

Calculating only the change in temperature due to changes in salinity, $\Delta\theta$ = -c_b * $\Delta\beta$ ~ -0.06 *- 0.004 ~ 2e-4 C which is similar to what is shown in the updated version of Fig. 2d. Thanks again for your thorough attention to our results.

Line by Line comments
* * *
34: state the salinity of sea water for comparison for those of us who do not deal with brine regularly

Good point. It now reads, "The salinity of the englacial brine feeding Blood Falls is approximately 125 psu (compared to ~35 psu for seawater) but the precise geometry…"

43: not sure what is meant by "edge dynamics"

We have changed this to read, "Channel size and evolution are also expected to differ as the result of fluid chemistry."

Table 1: for some variable their value is given (e.g. rho_i) but not for others (e.g. A). This should be consistent.

We state in the table caption that values of constants are specified, but A here is a function of temperature. Previously in line 93, we stated "We calculate A as a function of ice temperature using the Arrhenius relation and relevant calibrated values (Cuffey and Paterson, 2010, Eqns 3.35." We realize the necessary parameter values to calculate A were not all listed in Eqn 3.35 so we have added Eqn 3.36 to this reference.

99-101: Likely the biggest uncertainty in Manning's n is due to the great variability in its value (e.g. Pohle et al., 2022 find order of magnitude difference for one R-channel as it evolves). Maybe state something in this regard.

We have added in line 101, "Regardless of salinity, large uncertainties exist in Manning's formula even in fresh water systems due to the large variability in the value of Manning's friction factor (Pohle et al., 2022)."

Eq 7: I guess in this equation only the constant and linear term could be retained, the effects of the higher order terms are minimal for the salinities considered here.

Yes, true. We will delete the higher order terms.

143: use `\times` to typeset the "x" in scientific notation numbers

Good to know. Thanks.

Eq 9: it should be \sigma_w, the heat capacity of water (sorry, I miss-stated this in my last review); also in the summary of model equations

Thanks for pointing this out. We should have caught this after reviewing Eq 2.4 in Fowler (1999) upon your suggestion, but you are correct. It should be the heat capacity of the brine here, not ice.

The value of the specific heat capacity varies significantly as a function of salinity and temperature. We have now included this consideration in the model code and changed it in the manuscript. We have added the following to line 135 in the manuscript.

"Fluid properties such as the density, specific heat capacity, and the melting point of ice are functions of salinity ($\beta$ in practical salinity units [psu]). The specific heat capacity of brine is calculated with the salinity($\beta$ in practical salinity units [psu]) and temperature of the brine [C] in the lake following,
$\sigma$b = 4217.4 - 3.72 $\theta$_b - 7.64 $\beta$. using the first order terms from Eqn. A3.11 in Gill (1982)."

After this change, we reran all simulations to include the specific heat capacity of fluid, given the temperature and salinity. The results are similar, although there are now larger differences between saline and fresh water due to the larger specific heat of fluid compared to ice. This larger value of sigma further limits energy available for channel growth (in Eq. 10). We have edited all figures to include updated results. Slight adjustments have been made to the text to capture these changes. See below.

Changed lines 239-241 because the specific heat capacity of brine is constant along the channel… "As the walls of the channel melt and the brine is diluted, the density of the brine is not constant along the channel. We have accounted for this in our simulations, but neglecting these changes does not have a substantial influence on the results because the changes are very small (data not shown)."

Rewriting lines 228-230 based on the new results and for clarity, "With a salinity of 10 psu and an initial channel radius of 0.2 m, the lake drains substantially slower (≈ 10 days) compared to an initial channel radius of 0.3 m. For each 0.1 m increase in initial channel radius, the peak discharge increases non-linearly for both saline fluid and freshwater, but the rate of increase is less pronounced for saline fluid."

Replacing what was line 224, "For a freshwater lake, the peak discharge roughly triples when comparing $V_i = 1 × 10^5$ m^3 with $V_i = 5 × 10^5$ m^3 and $V_i = 5 × 10^5$ m^3 with $V_i = 1 × 10^6$ m^3, where as for a lake with a salinity of 10 psu the peak discharge approximately doubles."

Changed percentage slightly in line 334. "For a lake with a salinity of 125 psu, which is approximately the measured value at Blood Falls, and an initial brine temperature of −7.63◦ C, the peak velocity reached is 40% lower and the lake drains 9% slower than for a freshwater lake."

And in line 338 "22% higher than pure water…".

Also changed subscript in Table 1, summary of model equations, and appendix equations.

190: repetition

Deleted "Higher salinities require smaller time steps."

255: "evolving" -> "increasing"

Okay.

259: I'd rather use "reduce" than "limit" as the latter suggest a hard limit but it only reduces the effects. Also other occurrences, e.g. l295.

Agreed, changed this language everywhere.

287-289: state which assumption is more likely correct. Potentially cite, e.g. Röthlisberger here about his choice for the pressure melting point.

We have added "However, this assumption is likely unrealistic and differs from the convention in previous studies. In other models of subglacial channel flow that do not

explicitly include temperature, the temperature is set to follow the pressure melting point of the ice walls within the channel (e.g., Röthlisberger, 1972; Werder et al., 2013)."

Fig 4: plot 0psu also.

We choose not to plot 0 psu here because 0 psu already has a density of 1000 kg/m^3 and so there would be no difference between them. Removing 0 psu also changes the scale which makes the slight differences easier to visualize. We edited the caption to remove beta = 0 psu from the list.

Eq 15: Pretty sure this is wrong. In both denominators should be rho_s. This then leads to, for instance, a density of 1056kg/m3 for a SSC of 91g/l. Probably easiest to just correct the vector of SSCs in line 315 to give the stated rho-vector.

We agree the equation is wrong. We will change it to
ρc= ρw + ( ρs - ρw) ( SSC/ρs)

We believe this the equation you meant. We have changed the values of SSCs. We calculated the SSCs given the fluid densities we ran in the model. It now reads "We vary the suspended sediment concentrations from SSC = {0, 63, 125, 188, 251} g L^-1 to arrive at the combined fluid densities of rho_c = {1000, 1040, 1080, 1120, 1160} kg m^-3 to simulate different suspended sediment loading (for rho_s = 2760 kg m^-3)."

References

Fowler, A. C. (1999). "Breaking the Seal at Gr ́ımsv ̈otn, Iceland". In: Journal of Glaciology 45.151, pp. 506–516. doi: 10.3189/S0022143000001362.

Gill, A. E.: Atmosphere-ocean dynamics, Academic Press, 1982.

Pohle, A., Werder, M. A., Gräff, D., and Farinotti, D.: Characterising englacial R-channels using artificial moulins, J. of Glaciol, 68, 879–890, https://doi.org/10.1017/jog.2022.4, 2022.

R ̈othlisberger, H. (1972). "Water Pressure in Intra– and Subglacial Channels". In: Journal of Glaciology 11.62, pp. 177–203. doi: 10.3189/S0022143000022188.

Werder, M. A. et al. (2013). "Modeling Channelized and Distributed Subglacial Drainage in Two Dimensions". In: Journal of Geophysical Research: Earth Surface 118.4, pp. 2140–2158. doi: 10.1002/jgrf.20146.

---

## Author Response (AR4)

Dear Editor Nanna,

Thank you for your comments and your support in publishing our manuscript.

We have made the changes you have outlined as described.

On behalf of the authors,
Amy Jenson